# Characterization of Na⁺ currents regulating intrinsic excitability of optic tectal neurons

Adrian C Thompson, Carlos D Aizenman

Developing neurons adapt their intrinsic excitability to maintain stable output despite changing synaptic input. The mechanisms behind this process remain unclear. In this study, we examined *Xenopus* optic tectal neurons and found that the expressions of $Na_v1.1$ and $Na_v1.6$ voltage-gated Na⁺ channels are regulated during changes in intrinsic excitability, both during development and becsuse of changes in visual experience. Using whole-cell electrophysiology, we demonstrate the existence of distinct, fast, persistent, and resurgent Na⁺ currents in the tectum, and show that these Na⁺ currents are co-regulated with changes in $Na_v$ channel expression. Using antisense RNA to suppress the expression of specific $Na_v$ subunits, we found that up-regulation of $Na_v1.6$ expression, but not $Na_v1.1$, was necessary for experience-dependent increases in Na⁺ currents and intrinsic excitability. Furthermore, this regulation was also necessary for normal development of sensory guided behaviors. These data suggest that the regulation of Na⁺ currents through the modulation of $Na_v1.6$ expression, and to a lesser extent $Na_v1.1$, plays a crucial role in controlling the intrinsic excitability of tectal neurons and guiding normal development of the tectal circuitry.

## Introduction

Developing circuits have the ability to build specialized neural networks that can process and transmit information while the circuit itself continues to develop. As such, neurons in the developing nervous system face the challenging task of remaining adaptable to changes in circuitry and synaptic organization that occur during development or in response to sensory experiences, whereas at the same time, maintaining the ability to consistently respond to sensory inputs and process information (Marder & Goaillard, 2006; Turrigiano, 2012). Neurons achieve this balance by adjusting the number and strength of synaptic connections (Turrigiano, 2008), or by modifying their intrinsic excitability, which affects their firing rate (Turrigiano et al, 1994; Schulz, 2006). The distribution and function of voltage-gated ion channels, particularly voltage-gated sodium (Na⁺) channels, play a vital role in regulating intrinsic excitability (Raman et al,

1997; Rush et al, 2005), as they are strategically positioned to influence processes like action potential waveform and threshold (Blair & Bean, 2002; Van Wart & Matthews, 2006). Although it has been established that changes in Na⁺ current amplitude and kinetics can modulate intrinsic excitability (Raman et al, 1997; Rush et al, 2007; Khaliq & Bean, 2010), the specific molecular mechanisms for the dynamic changes in Na⁺ currents, that cause homeostatic changes in neuronal excitability, remain elusive.

*Xenopus laevis* tadpoles perform visually guided behaviors even as the optic tectum, the principal midbrain structure for sensory integration, continues to develop. Experience-dependent development of this sensory circuit relies on the tight regulation of intrinsic excitability (Dong & Aizenman, 2012), with changes in excitability correlated with changes in the amplitude of Na⁺ currents (Aizenman et al, 2003; Pratt & Aizenman, 2007; Ciarleglio et al, 2015). However, little is known about what mechanism is responsible for changes in Na⁺ currents, nor of the molecular underpinnings of voltage-gated currents in this system. Knowledge of these mechanisms can provide important insights as to how neurons regulate excitability to ensure that they continue to function correctly even as the wider circuit continues to develop and undergo structural and functional rearrangement.

Voltage-gated Na⁺ channels mediate distinct fast, persistent, and resurgent Na⁺ currents that display characteristic time scales, voltage dependencies, and gating properties; all of which can influence neuronal excitability. Fast and persistent Na⁺ currents have been identified in *Xenopus* tectal neurons (Aizenman et al, 2003; Hamodi et al, 2016), with fast Na⁺ currents known to be regulated with changes in neuronal excitability across development and with experience-dependent changes in synaptic strength (Aizenman et al, 2003; Pratt & Aizenman, 2007; Hamodi et al, 2016). It is not known whether resurgent currents exist in this developing system, or whether they can also be regulated by experience.

In this study, our objective was to investigate the molecular mechanisms through which neurons in the optic tectum adjust their intrinsic excitability during both circuit development and with enhanced sensory experience. Understanding these mechanisms is crucial for comprehending how retinotectal circuits are formed. We found that tectal neurons homeostatically adapt their intrinsic excitability during development and in response to visual experience by altering the amplitude of specific Na⁺ currents, namely

Department of Neuroscience, Brown University, Providence, RI, USA

Correspondence: Carlos_Aizenman@Brown.edu

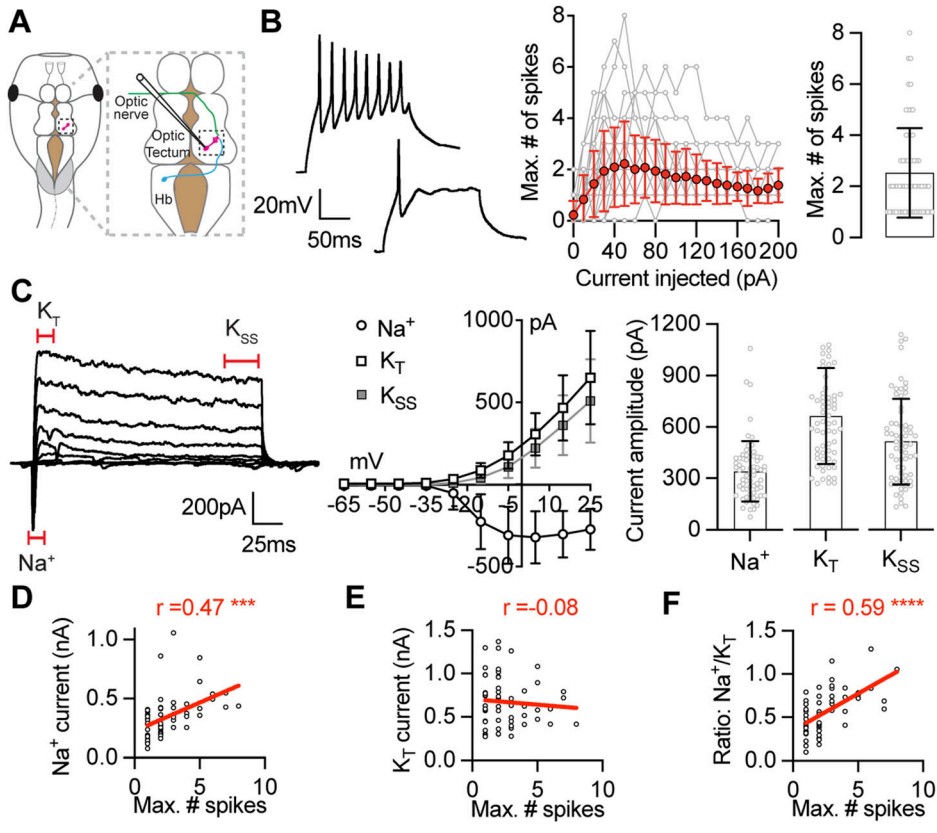

**Figure 1. Intrinsic excitability of *Xenopus* tectal neurons is correlated with the amplitude of voltage-gated Na⁺ currents.**

**(A)** Diagram shows a *Xenopus* tadpole illustrating whole-cell recordings from a neuron of the optic tectum that receives innervation from visual and mechanosensory inputs. **(B)** *Left*: example current-clamp recordings showing the spiking response of two stage 49 tectal neurons held at −65 mV to a 50 pA current injection, illustrating the range of responses observed. *Middle*: plot shows the number of spikes elicited in response to 0–200 pA current injections for all cells analyzed (grey), and the average response (red, mean ± SD). *Right*: maximum number of spikes (median = 2 spikes, IQR 1–3 spikes, $n$ = 61 cells). **(C)** *Left*: example voltage-clamp recording from a tectal neuron held at −65 mV in response to a series of depolarizing steps (−65 to +25 mV). Recording is leak subtracted to show only active currents. *Middle*: I–V plot shows average current amplitude for the Na⁺, the transient K⁺ ($K_T$) and the steady state K⁺ currents ($K_{SS}$). *Right*: peak current amplitudes (Na⁺: 341.4 ± 176.1 pA; $K_T$: 666.9 ± 281.0 pA; $K_{SS}$: 517.1 ± 251.2 pA; $n$ = 61 cells). **(D, E, F)** Plots show Pearson correlations between intrinsic excitability (max. number of spikes) and (D) the amplitude of the Na⁺ current, (E) the transient K⁺ current, and (F) the ratio of the Na⁺ current to transient K⁺ current. r values are Pearson correlation coefficients (***$P$ = 0.0002; ****$P$ < 0.0001). The complete Pearson correlations matrix showing the relationships between all biophysical properties measured is presented in Fig S1.

Source data are available for this figure.

fast, persistent, and resurgent Na⁺ currents. Critically, we show that this adaptation required changes in expression of Na⁺ channel subtype Na$_v$1.6, which is a requirement for sensory experience-dependent homeostatic increases in Na⁺ current amplitude and intrinsic excitability. We further extend these findings to show that this mechanism is critical for the functional development of the retinotectal circuity, as dysregulation of Na$_v$1.6 channel expression during a key period of development, causes deficits in behaviors that depend on visual and multisensory processing. Overall, these findings highlight the critical role that dynamic regulation of Na⁺ channel gene expression plays in the homoeostatic regulation of neuronal excitability, and the importance of this process for normal circuit development. This mechanism enhances our understanding of the molecular factors influencing excitability in the developing nervous system and underscores the need to better understand the role of Na⁺ channel subtype-specific current adaptation in regulating circuit formation during nervous system development.

## Results

### The intrinsic excitability of *Xenopus* tectal neurons is correlated with the amplitude of voltage-gated Na⁺ currents

Principal tectal neurons, which are the main recipients of retinal input in the developing visual circuit of tadpoles, must adapt their intrinsic excitability to maintain neuronal function in response to

changes in synaptic input as a result of both developmental circuit reorganization and sensory experience (Aizenman et al, 2003; Pratt & Aizenman, 2007; Ciarleglio et al, 2015; Busch & Khakhalin, 2019). Although studies strongly implicate the regulation of Na⁺ currents with changes in tectal neuron excitability (Aizenman et al, 2003; Pratt & Aizenman, 2007; Ciarleglio et al, 2015), there is also evidence that the regulation of K⁺ currents plays a role in this process (Hamodi & Pratt, 2014; Ciarleglio et al, 2015). As such, the molecular mechanisms by which Na⁺ currents are regulated to control intrinsic excitability, and the degree to which regulation of K⁺ currents contributes to this control of excitability, are not fully understood.

To examine the relationship between intrinsic excitability of tectal neurons and the regulation of voltage-gated Na⁺ and K⁺ currents, we performed whole-cell recordings on 61 deep-layer principal tectal neurons from tadpoles at developmental stage 49 using a whole brain ex vivo preparation (Fig 1A). We selected stage 49 tadpoles because the biophysical properties of their tectal neurons is heterogenous at this developmental stage, with most neurons exhibiting low intrinsic excitability but with some more highly excitable neurons still observed (Ciarleglio et al, 2015). As expected, we observed a broad range of values in the maximum number of spikes that tectal neurons could generate (1–8), with most neurons spiking once or twice (Fig 1B). For each neuron, we also measured the peak current amplitudes for the voltage-gated Na⁺ current, the transient K⁺ current ($K_T$), and the steady-state K⁺ current ($K_{SS}$) (Fig 1C). Importantly, these biophysical properties matched previous reports for stage 49 tectal neurons (Aizenman

et al, 2003; Pratt & Aizenman, 2007; Ciarleglio et al, 2015; James et al, 2015) and showed a wide range of excitability levels which allowed us to investigate the cellular basis for heterogeneity of intrinsic excitability within tectal neurons.

To measure the relationship between intrinsic excitability and other intrinsic properties of tectal neurons, we performed a multivariate analysis and calculated Pearson pairwise correlations between each of the biophysical properties measured (the complete correlation matrix is shown in Fig S1). When comparing voltage clamp currents with the maximum number of spikes, we found a strong correlation with the peak amplitude of the $Na^+$ current (R 0.47; Fig 1D), whereas the peak amplitude of the transient $K^+$ current was weakly correlated with the maximum number of spikes (R –0.08; Fig 1E). Accordingly, we also found a strong correlation between the maximum number of spikes and the $Na^+$ to transient $K^+$ current ratio (R 0.59; Fig 1F). We restricted our measurements to peak current amplitude, and not activation and inactivation kinetics because of possible space-clamp issues, consistent with previous studies (Aizenman et al, 2003; Pratt & Aizenman, 2007). These data indicate that the regulation of $Na^+$ currents is an important mechanism to control the intrinsic excitability of tectal neurons, with an increased ratio of $Na^+$ to $K^+$ currents present in more excitable tectal neurons. In further support of this idea, we measured significant correlations between the peak amplitude of the $Na^+$ current and characteristics of the first spike including rate of rise and spike width; but little correlation between characteristics of the first spike and the amplitude of $K^+$ currents (Fig S1). Taken together, these data provide further evidence that $Na^+$ currents are a key determinate of the intrinsic excitability of tectal neurons.

## Expression of $Na^+$ channel subtypes are differentially regulated in the *Xenopus* optic tectum during key developmental time windows and in response to changes in network activity

How are voltage-gated $Na^+$ currents regulated to control intrinsic excitability? $Na^+$ channels are comprised of alpha and beta subunits, the alternate expression of which confers $Na^+$ channels with distinct cellular expression profiles, subcellular localization, conduction properties, and responses to $Na^+$ channel activators and inhibitors (Savio-Galimberti et al, 2012). The alpha subunits ($Na_v1.1$–$Na_v1.9$) are the pore-forming components of $Na^+$ channels, whereas the accessory beta subunits ($Na_v1\beta$–$Na_v4\beta$) can regulate expression, cellular localization, and gating properties of $Na^+$ channels (O'Malley & Isom, 2015). A search of the *X. laevis* genome (Xenbase: *X. laevis* version 9.2 on JBrowse) revealed that *X. laevis* expresses alpha subunit genes encoding for the brain-expressed voltage-gated $Na^+$ channel subtypes $Na_v1.1$, $Na_v1.2$, and $Na_v1.6$. *Xenopus* also expresses the accessory beta subunit $Na_v4\beta$ that has a well-described role in regulating neuronal excitability by mediating resurgent $Na^+$ currents (Bant & Raman, 2010). If the regulation of $Na^+$ channel expression is a mechanism to control voltage-gated $Na^+$ currents and intrinsic excitability, then we would predict that the expression of individual $Na^+$ channel genes would correlate with changes in the intrinsic excitability of tectal neurons. Because the intrinsic excitability of tectal neurons is regulated across tectal circuit development and increased in response to a short-term patterned sensory experience (Aizenman et al, 2003; Pratt &

Aizenman, 2007; Ciarleglio et al, 2015), we used qRT-PCR to quantify the expression of $Na_v1.1$, $Na_v1.2$, $Na_v1.6$, and $Na_v4\beta$ subunits in the optic tectum in these conditions.

As the tectal circuitry matures between developmental stages 42 and 49, the intrinsic excitability of tectal neurons peaks at stage 46 congruent with an overall increase in the level of excitatory synaptic input at this stage of circuit development (Fig 2A) (Pratt & Aizenman, 2007). If the expression levels of individual $Na^+$ channel genes correlate with changes in neuronal intrinsic excitability, we would predict increased expression at developmental stage 46. Interestingly, we found that the expression levels of $Na_v1.1$ and $Na_v1.6$ in the optic tectum were regulated developmentally, but with differing developmental expression profiles (Fig 2B and Table S1). We found that the expression levels of $Na_v1.1$ increased with development. We measured a ~twofold increase in $Na_v1.1$ expression levels between stage 42 and stage 49 (st 42: 1.00 ± 0.12, st 49: 1.84 ± 0.31; *P* = 0.0151), but no significant change between stages 42 and stage 46 (st 46: 1.57 ± 0.25; *P* = 0.0535) or between stage 46 and stage 49 (*P* = 0.2106). Although this increase in $Na_v1.1$ levels may signal maturation of neurons, it does not correlate with developmental changes in intrinsic excitability. In contrast, $Na_v1.6$ expression levels peaked at developmental stage 46, when intrinsic excitability is highest. Expression of $Na_v1.6$ was increased ~twofold between stage 42 and stage 46 (st 42: 1.00 ± 0.09, st 46: 1.71 ± 0.22; *P* = 0.0350), with the level of expression at stage 49 being not significantly different from stages 46 (st 49: 1.40 ± 0.32; *P* = 0.1932) or stages 42 (*P* = 0.1932). Unlike the developmental regulation of $Na_v1.1$ and $Na_v1.6$ channels, the expression levels of $Na_v1.2$ and $Na_v4\beta$ remained largely unchanged across development (Fig 2B). These data suggest that $Na_v1.1$ channel expression increases with the functional maturation of tectal neurons, whereas the expression of $Na_v1.6$ channels more closely matches developmental changes in intrinsic excitability.

To test whether observed changes in $Na^+$ channel expression levels were simply a function of tectal neuron development, or whether changes in $Na^+$ channel expression represents a mechanism for the homeostatic regulation of in tectal neuron excitability, we quantified the expression of $Na^+$ channel genes in the optic tectum of stage 49 tadpoles exposed to 4 h of enhanced visual stimulation (EVS). EVS is a well-established method for triggering increased intrinsic excitability of tectal neurons as a homeostatic response to decreased excitatory synaptic drive (Fig 2A) (Aizenman et al, 2003; Ciarleglio et al, 2015). Critically, we found that $Na_v1.6$ expression levels, and to a lesser extent $Na_v1.1$ expression levels, were increased in response to EVS (Fig 2C). The level of $Na_v1.6$ expression was increased ~fourfold in the optic tectum of tadpoles exposed to EVS compared with naïve stage 49 controls (Ctrl: 1.00 ± 0.18, EVS: 3.80 ± 1.13; *P* = 0.0135). We also found a ~threefold increase in the expression of $Na_v4\beta$ (Ctrl: 1.00 ± 0.07, EVS: 3.21 ± 1.31; *P* = 0.0434), whereas a more modest ~twofold increase in the level of $Na_v1.1$ expression was observed in the optic tectum of EVS exposed tadpoles (Ctrl: 1.00 ± 0.23, EVS: 1.71 ± 0.13; *P* = 0.0108), whereas $Na_v1.2$ expression levels were unchanged between control tadpoles and EVS-exposed tadpoles (Ctrl: 1.00 ± 0.13, EVS: 1.00 ± 0.09; *P* = 0.9460). Given the well-established link between $Na_v1.6$ and $Na_v4\beta$ expression and the regulation of resurgent $Na^+$ currents and intrinsic excitability (Grieco et al, 2005; Bant & Raman, 2010; Lewis &

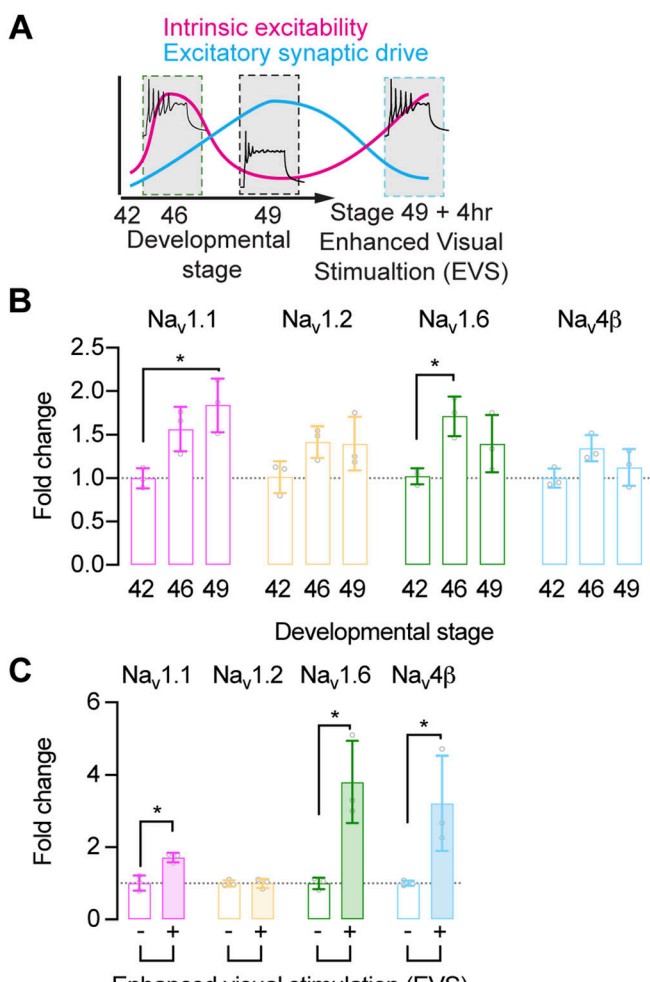

Figure 2. The expression of Na⁺ channel genes is regulated with developmental and homeostatic changes in neuronal intrinsic excitability.
**(A)** Schematic illustrates how tectal neurons homeostatically adapt intrinsic excitability in response to changing excitatory synaptic drive across development and in response to 4 h exposure to enhanced visual stimulation (EVS) to maintain a broad dynamic range and, thereby, conserve input–output function as the tectal circuitry changes (Aizenman et al, 2003; Pratt & Aizenman, 2007; Ciarleglio et al, 2015). **(B)** Expression levels of selected brain-expressed voltage-gated Na⁺ channel alpha and beta subunits in the optic tectum at key development stages. Developmental stages were selected to sample immature tectal neurons with low excitability (stage 42), highly excitable tectal neurons undergoing experience-dependent circuit remodeling (stage 46), and mature tectal neurons with low excitability (stage 49). The expression of Na⁺ channel alpha and beta subunits genes was normalized to a housekeeper (RSP13), before determining the fold change in expression from developmental stage 42 (Na$_V$1.1 stage 42 versus stage 49, $P = 0.0151$; Na$_V$1.6: stage 42 versus stage 46, $P = 0.0350$. $N = 3$ experiments with RNA isolated from 10 tadpoles). See Table S1 for comparisons of expression levels between all developmental stages. **(C)** Expression levels of selected Na⁺ channel alpha and beta subunits in the optic tectum of stage 49 tadpoles with or without 4 h exposure to EVS, which triggers a homeostatic increase in neuronal excitability. Expression is shown as the fold change in normalized Na⁺ channel alpha and beta subunit gene expression compared with naïve stage 49 control tadpoles (Ctrl versus EVS: Na$_V$1.1, $P = 0.0108$; Na$_V$1.6, $P = 0.0135$; Na$_V$4$\beta$, $P = 0.0434$. $N = 3$ experiments with RNA isolated from 10 tadpoles).

Raman, 2011), these data provide evidence that tectal neurons homeostatically control their intrinsic excitability by regulating the amplitude of Na⁺ currents through changes in Na$_V$1.6 gene

expression levels, and perhaps via the regulation of a resurgent Na⁺ current. Our data also show that up-regulation of Na$_V$1.1 expression contributes to the increased amplitude of Na⁺ currents as intrinsic excitability is homeostatically increased. However, the modest up-regulation of Na$_V$1.1 compared with Na$_V$1.6, and the fact that Na$_V$1.1 expression levels were most elevated at stage 49 when tectal cells are less excitable, would indicate that their role in regulating excitability may be less important compared with Na$_V$1.6 channels.

### Tectal neurons express distinct fast and persistent Na⁺ currents that are regulated with homeostatic changes in intrinsic excitability across development and in response to EVS

Voltage-gated Na⁺ channels carry both fast and persistent Na⁺ currents, which have distinct roles in action potential generation and the repetitive firing of neurons (French et al, 1990; Raman et al, 1997; Magistretti et al, 2006). The fast Na⁺ current mediates the upstroke of action potentials before becoming rapidly inactivated (Hodgkin & Huxley, 1952; Stuart & Sakmann, 1994; Martina & Jonas, 1997). When Na⁺ channels fail to fully inactivate, even with pro-longed depolarization, the resulting sustained current is termed a persistent Na⁺ current (French et al, 1990). Because the persistent Na⁺ current is a depolarizing current that occurs at a subthreshold voltage range, it can amplify a neuron's response to synaptic input and enhance the neurons capacity to fire repetitively (Bant & Raman, 2010; Chen et al, 2011). We therefore hypothesized that tectal neurons modulate fast and persistent Na⁺ currents to achieve homeostatic changes in intrinsic excitability. Although fast Na⁺ currents are known to be modulated with changes in excitability in *Xenopus* tectal neurons (Aizenman et al, 2003; Pratt & Aizenman, 2007), the relative contribution of fast and persistent Na⁺ currents to changes in excitability has not been directly examined.

To determine whether fast and persistent Na⁺ currents are regulated with changes in intrinsic excitability that occur across development (refer to diagram in Fig 2A), we performed whole-cell recordings on tectal neurons from tadpoles at developmental stages 42, 46, and 49. Na⁺ currents were isolated in voltage-clamp recordings using a Tris-based internal saline solution to block outward K⁺ currents (Aizenman et al, 2003), which revealed distinct fast and persistent voltage-gated Na⁺ currents (Fig 3A). When we compared peak amplitudes of the fast and persistent Na⁺ currents across development, we observed a transient peak for fast and persistent Na⁺ currents at stage 46 (Fig 3B and C), congruent with the increased intrinsic excitability at this developmental stage (Pratt & Aizenman, 2007), and with changes in Na⁺ channel expression observed in this study.

We also recorded Na⁺ currents from stage 49 tectal neurons after the exposure of tadpoles to 4 h of EVS to trigger a homeostatic increase in the excitability of tectal neurons. When we compared the amplitude of the fast and persistent Na⁺ currents in tectal neurons of EVS-exposed stage 49 tadpoles with naïve stage 49 tadpoles, we found a significant increase in both currents (Fig 3B and C). Thus, we find a correlation between changes in excitability, Na⁺ current amplitude, and changes in Na⁺ channel expression levels.

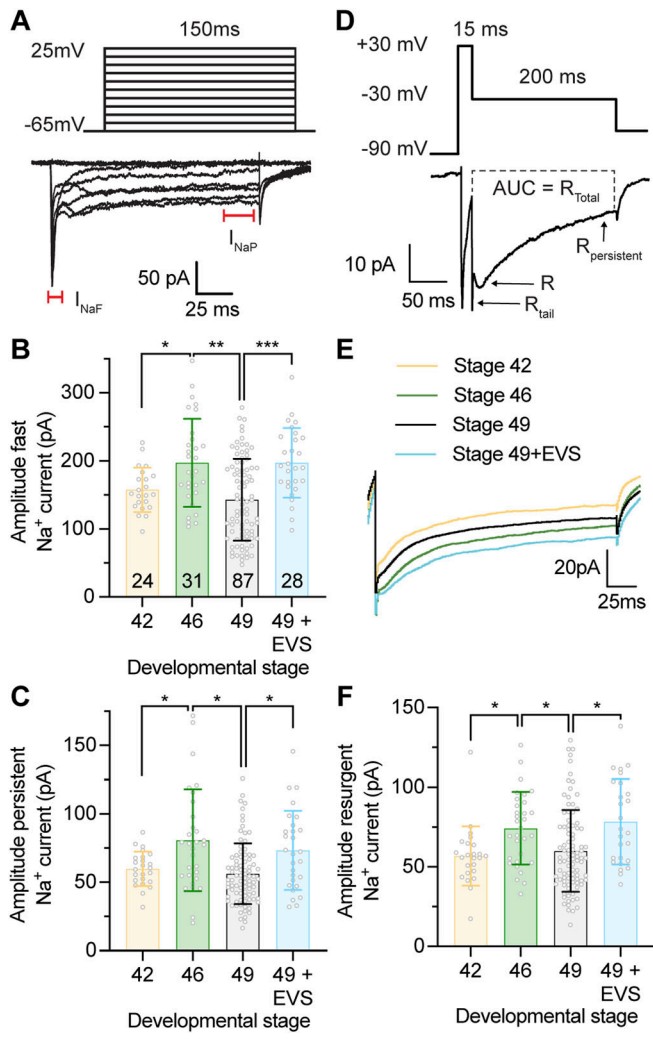

**Figure 3. Tectal neurons express fast and persistent voltage-gated Na+ currents that are regulated with developmental and homeostatic changes in intrinsic excitability.**
**(A)** Example voltage-clamp recordings from a tectal neuron held at −65 mV in response to a series of 150 ms depolarizing steps (−65 to 25 mV) using a Tris-based internal saline solution, which blocks outward K+ currents to reveal distinct fast and persistent Na+ currents. Recordings are leak subtracted to show only active currents. Fast ($I_{NaF}$) and persistent ($I_{NaP}$) Na+ currents are indicated on the recording. **(B, C)** Quantifications of peak amplitudes for the fast and persistent Na+ currents. **(B)** Fast Na+ current ([in pA] Stage 42: 157.7 ± 32.5; Stage 46: 197.3 ± 64.4; Stage 49: 143.1 ± 60.3; Stage 49 + enhanced visual stimulation [EVS]: 197.4 ± 50.9). **(C)** Persistent Na+ current ([in pA] Stage 42: 59.8 ± 12.5; Stage 46: 80.7 ± 37.3; Stage 49: 56.3 ± 22.3; Stage 49 + EVS: 73.4 ± 29.0). **(D)** Example resurgent current recording from a tectal neuron held at −65 mV that was hyperpolarized to −90 mV for 500 ms before stepping to +30 mV for 15 ms to open voltage-gated Na+ channels. Resurgent Na+ currents were then recorded by a repolarizing step to −30 mV for 200 ms, which revealed distinct tail resurgent ($R_{Tail}$), resurgent (R) and persistent resurgent ($R_{Persistent}$) currents. Recordings were obtained using a Tris-based internal saline solution to block outward K+ currents, and leak subtracted to show only active currents. **(E)** Averaged resurgent current traces obtained from tectal neurons between developmental stages 42–49, and at stage 49 after exposure to 4 h of EVS. **(F)** Peak resurgent current amplitude across development and in response to 4 h of EVS ([in pA] Stage 42: 55.9 ± 18.6; Stage 46: 74.3 ± 22.8; Stage 49: 60.0 ± 25.7; Stage 49 + EVS: 78.4 ± 26.9). Groups were compared using a Welch's ANOVA test with Dunnett T3 test for multiple comparisons. **(D)** n values are shown in (D).
Source data are available for this figure.

## Tectal neurons express resurgent Na+ currents that are regulated with homeostatic changes in intrinsic excitability across development and in response to EVS

Voltage-gated Na+ channels can also generate a resurgent Na+ current that is known to facilitate repetitive firing in many types of neurons (Raman et al, 1997; Cummins et al, 2005; Theile & Cummins, 2011; Browne et al, 2017). The resurgent Na+ current is a sub-threshold depolarizing current that results from a distinctive gating mechanism, typically by open-channel block mediated by accessory proteins including $Na_V4\beta$ (Grieco et al, 2005; Bant & Raman, 2010). Open-channel block is relieved at subthreshold potentials, generating a resurgent current that promotes firing. Because we had observed that exposure to EVS triggered an up-regulation of $Na_V4\beta$ in the optic tectum, we asked if tectal neurons express a resurgent Na+ current and whether this resurgent Na+ current is regulated with homeostatic changes in the excitability of tectal neurons.

We followed established protocols for measuring resurgent currents (Raman et al, 1997). We first hyperpolarized the neuron to −90 mV to shift inactivated Na+ channels into a closed state, and then briefly depolarized to 30 mV for 15 ms to open Na+ channels, before performing a repolarizing step to −30 mV for 200 ms to generate the resurgent Na+ currents (Fig 3D). Significantly, we detected a resurgent Na+ current with distinct components (see labelled example trace in Fig 3D). We observed a fast tail current ($R_{Tail}$) that occurred ~1 ms after cells were repolarized to −30 mV (rise time [0–100%] = 0.92 ± 0.39 ms; n = 156 cells across all de-velopmental stages). We termed this current a tail resurgent Na+ current as the amplitude of the tail current had a near linear voltage dependency as the voltage of the resurgent step was changed from −80 to 20 mV (Fig S2A and B), and was attenuated as the time of the depolarizing step increased (Fig S2C and D), which is consistent with characteristics of tail Na+ currents (Raman et al, 1997). We also observed a resurgent Na+ current (R) that peaked at ~7.5 ms (rise time [0–100%] = 7.50 ± 9.64 ms) before decaying to a persistent state with a half-life of ~24 ms (decay time [100–50%] = 23.99 ± 13.64 ms). Significantly, the resurgent Na+ current peaked at −60 mV when the repolarization voltage was stepped from −90 to +30 mV (Fig S2A and B), decreased as the time of the depolarizing step increased (Fig S2C and D), and increased as the voltage of the depolarizing step was increased from 0 to +30 mV (Fig S2E and F), consistent with what has previously been described for resurgent Na+ currents (Patel et al, 2016). Lastly, we identified a persistent component of the resurgent Na+ current ($R_{Persistent}$) that showed voltage kinetics similar to the persistent Na+ current (Fig S2A and B). The resurgent Na+ current most closely resembled resurgent cur-rents initially described in rodent Purkinje neurons (Raman et al, 1997; Raman & Bean, 2001), and it is the first description of a re-surgent Na+ current in *Xenopus* tectal neurons.

If the regulation of a resurgent Na+ current is a mechanism used by tectal neurons to regulate intrinsic excitability, then we would predict that the resurgent current would be regulated with ho-meostatic changes in intrinsic excitability across development and with exposure to visual stimulation. Consistent with what we had observed for the fast and persistent Na+ currents, we found that the

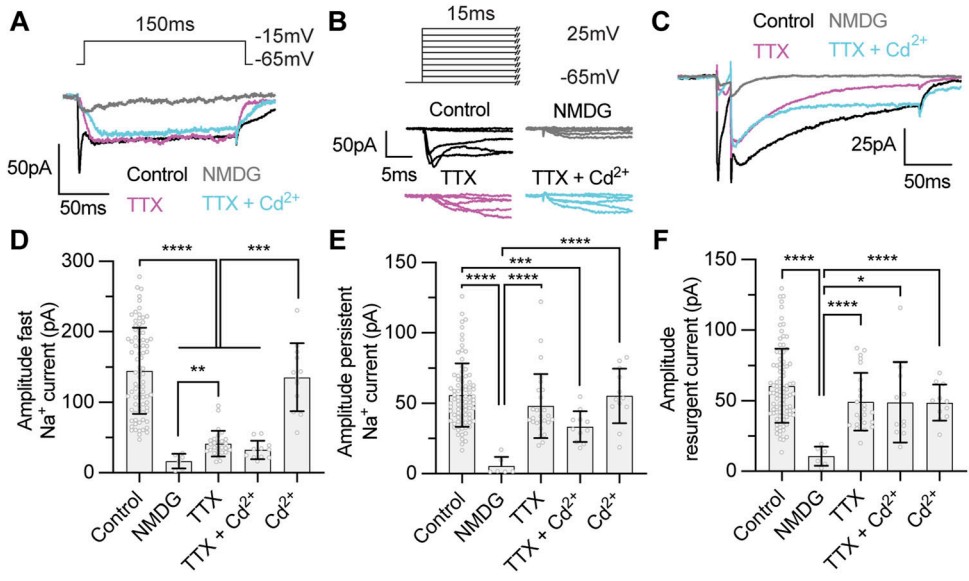

**Figure 4. The persistent and resurgent Na⁺ currents, but not the fast Na⁺ current, are insensitive to tetrodotoxin (TTX).**
**(A, B, C)** Example voltage-clamp recordings from tectal neurons with a Tris-based internal solution to isolate Na⁺ currents from stage 49 tectal neurons in control conditions (black), in zero Na⁺ external solution (NMDG) to block all inward Na⁺ currents (grey), in 1 μM TTX to block TTX-sensitive Na⁺ currents (magenta), or in 100 nM Cd²⁺ to block Ca²⁺ currents (cyan). **(A)** Example recordings show fast and persistent from a single depolarizing step to –15 mV for each experimental group. **(A, B)** Magnification of the initial 15 ms of the recordings shown in (A) to a series of voltage steps (–65 to +25 mV) to highlight the effect of each treatment on the fast Na⁺ current. Note that TTX attenuates fast but not persistent Na⁺ currents, whereas blocking all Na⁺ influx by replacing external Na⁺ with NMDG attenuates both fast and persistent currents to reveal a small, presumptive Ca²⁺ current. **(C)** Example recordings illustrating the effect of each condition on the resurgent Na⁺ current.
**(D, E, F)** Quantification of peak amplitudes for the (D) fast, (E) persistent, and (F) resurgent Na⁺ currents for each experimental group. Groups were compared using a Welch's ANOVA test with Dunnett T3 test for multiple comparisons. *n* values for (D, E, F) were 87 stage 49 controls, 15 NMDG, 27 TTX, 11 TTX + Cd²⁺, and 12 Cd²⁺. Values and comparisons are shown in Table S2.
Source data are available for this figure.

resurgent Na⁺ current transiently peaked at developmental stage 46 and was increased in response to EVS (Fig 3E and F). Crucially, this regulation of resurgent Na⁺ current with changes in excitability was not observed for the tail or persistent resurgent Na⁺ current (Fig S3A–C), providing evidence for our identification of the resurgent Na⁺ current. These data show that changes in the resurgent Na⁺ current correlate with changes in intrinsic excitability, suggesting that tectal neurons regulate fast, persistent, and resurgent Na⁺ currents together to control homeostatic changes in intrinsic excitability.

### Persistent and resurgent Na⁺ currents, but not the fast Na⁺ current, are insensitive to tetrodotoxin (TTX)

We next performed a series of experiments utilizing either ion substitution or specific channel inhibitors to further characterize the ionic permeability and channel types that contribute to the fast, persistent, and resurgent Na⁺ currents observed in tectal neurons.

To confirm that persistent and resurgent Na⁺ currents identified in this study were indeed the result of Na⁺ influx, we performed an ion substitution experiment by recording voltage-clamp currents when extracellular Na⁺ was replaced with NMDG. NMDG is an impermeant organic monovalent cation that abolishes inward Na⁺ currents (Blair & Bean, 2002) (Fig 4A–D). When we recorded from tectal neurons in NMDG external using a Tris-based internal solution (compare black and grey traces in Fig 4A–C), we found that blocking Na⁺ influx abolished the fast, persistent and resurgent Na⁺ currents (Fig 4D–F and Table S2). Abolishing Na⁺ currents revealed a small presumptive Ca²⁺ current as previously described (Hamodi & Pratt, 2014). To show that the effect of NMDG external was specific to

Na⁺ currents, we also recorded voltage-clamp currents using a K⁺-based internal (Fig S4), which confirmed that NMDG specifically abolishes Na⁺ currents. Furthermore, to exclude the possibility that the observed persistent and resurgent currents are the result of inward flux of K⁺ ions, we measured Na⁺ currents when K⁺ channels were blocked by performing recordings with a TEA-based external saline. Not unsurprisingly, we observed no effect of blocking K⁺ influx on Na⁺ currents (Table S2). These data suggest that the persistent and resurgent Na⁺ currents identified in this study are carried by voltage-gated Na⁺ channels.

TTX-resistant Na⁺ currents have been observed in neurons of anuran species (Campbell, 1992a, 1992b; Kobayashi et al, 1993, 1996). Although the fast Na⁺ current in *Xenopus* tectal neurons is sensitive to the Na⁺ channel blocker TTX (Aizenman et al, 2003), it remained to be determined whether persistent and resurgent Na⁺ currents are sensitive to TTX. Because mammalian Na$_v$1.1, Na$_v$1.2, and Na$_v$1.6 channel subtypes are TTX-sensitive, we predicted that TTX would also abolish the persistent and resurgent currents in tectal neurons. However, when we measured Na⁺ currents in the presence of 1 μM TTX (magenta traces in Fig 4A–C), we found that TTX attenuated the fast Na⁺ current as expected; however, surprisingly, there was no significant effect of TTX on the amplitude of persistent or resurgent Na⁺ currents (Fig 4D–F and Table S2). Moreover, the persistent and resurgent Na⁺ currents remained after the concentration of TTX was increased to 30 μM (Table S2), suggesting that the persistent and resurgent Na⁺ currents are largely TTX-insensitive in *Xenopus* tectal neurons.

As we had observed no effect of TTX on the persistent and resurgent Na⁺ currents, we next tested whether the less-specific voltage-gated Na⁺ channels blocker lidocaine attenuates persistent and resurgent Na⁺ currents (Fig S5A and B). Because lidocaine

binds Na$^+$ channels in the inactivated state, we measured Na$^+$ currents after a 10 s depolarizing step to 0 mV. When we measured Na$^+$ currents in the presence of 1 μM lidocaine, we found a near abolishment of fast, persistent, and resurgent Na$^+$ currents (Table S2). These data provide further evidence to suggest that persistent and resurgent Na$^+$ currents in *Xenopus* tectal neurons are carried by TTX-resistant voltage-gated Na$^+$ channels.

We had observed a small, presumptive Ca$^{2+}$ current in the absence of extracellular Na$^+$. Therefore, we examined whether the TTX-insensitive persistent and resurgent Na$^+$ currents could be mediated by Ca$^{2+}$ influx via voltage-gated Ca$^{2+}$ channels. To test this, we recorded currents in the nonspecific voltage-gated Ca$^{2+}$ channels blocker Cd$^{2+}$ in the presence or absence of TTX (cyan traces in in Fig 4A–C). Crucially, we observed no effect of blocking voltage-gated Ca$^{2+}$ channels on the amplitude of the fast, persistent or resurgent Na$^+$ current (Fig 4D–F and Table S2). These findings show that Ca$^{2+}$ influx contributes little to the persistent and resurgent current, consistent with the idea that the TTX-insensitive component of the persistent and resurgent Na$^+$ currents being mediated by Na$^+$ influx via voltage-gated Na$^+$ channels. Taken together, these experiments confirm that tectal neurons express a TTX-sensitive fast Na$^+$ current and extend this finding to demonstrate the presence of a TTX-insensitive component of the persistent and resurgent Na$^+$ currents.

### Inhibition of Na$^+$ channel subtype Na$_v$1.6 attenuates resurgent Na$^+$ currents and decreases the intrinsic excitability of tectal neurons

To understand the mechanisms by which Na$^+$ currents are dynamically regulated with changes in the excitability of tectal neurons, we sought to determine the molecular composition of the fast, persistent, and resurgent Na$^+$ currents. Our expression analysis had suggested that Na$_v$1.1 and Na$_v$1.6 channels may contribute to the regulation of Na$^+$ currents and intrinsic excitability, whereas Na$_v$1.2 channels s did not appear to be regulated with intrinsic excitability. Although Na$_v$1.6 is TTX-sensitive in mammalian neurons (Lee & Ruben, 2008), it is evolutionarily distinct from Na$_v$1.1 and Na$_v$1.2 (Zakon, 2012), and has been shown to mediate persistent and resurgent Na$^+$ currents (Raman et al, 1997; Bant & Raman, 2010; Patel et al, 2015). Furthermore, *Xenopus* and other anuran species have a tyrosine to phenylalanine substitution at the same residue within the P-loop of domain I of Na$_v$1.6 that has been shown to be critical for TTX binding in the channel pore (Herzog et al, 2003), which may confer *Xenopus* Na$_v$1.6 with distinct TTX sensitivity. We therefore hypothesized that Na$_v$1.6 could be a candidate to mediate TTX-insensitive components of the persistent and resurgent Na$^+$ currents in *Xenopus* tectal neurons.

To examine the contribution of Na$_v$1.6 channels to the regulation of persistent and resurgent Na$^+$ currents and intrinsic excitability in tectal neurons, we performed whole-cell recordings that measured Na$^+$ currents and intrinsic excitability in the presence of the specific Na$_v$1.6 inhibitor MV1312 (Fig 5). MV1312 shows five to sixfold sensitivity for Na$_v$1.6 over Na$_v$1.1, and has been shown to rescue seizure behavior in a zebrafish model of Dravet syndrome where a loss-of-function Na$_v$1.1 function results in overexpression of Na$_v$1.6 and epileptogenesis (Weuring et al, 2020). MV1312 is predicted to

bind Na$^+$ channels similar to lidocaine, therefore we performed whole-cell recordings with Tris-based internal solution and acutely washed in 5 μM MV1312 with recordings preceded by a 5-s depolarizing step to 0 mV to open Na$^+$ channels (Fig 5A). We found that blocking Na$_v$1.6 channels caused a significant attenuation of all Na$^+$ currents, with the largest effect observed for the resurgent Na$^+$ current (Fig 5B–D). These findings suggest that MV1312 blocks Na$^+$ channels in the inactivated state with slow kinetics, and provide evidence that Na$_v$1.6 channels are a major contributor to fast, persistent, and resurgent Na$^+$ currents in *Xenopus* tectal neurons.

If Na$_v$1.6-mediated Na$^+$ currents are important regulators of intrinsic excitability in tectal neurons, then inhibition of Na$_v$1.6 channels would be predicted to reduce neuronal excitability. To measure the effect of Na$_v$1.6 channel inhibition on the intrinsic excitability of tectal neurons, we performed whole-cell recordings with a K$^+$-based internal solution in the presence of 5 μM MV1312. Because tectal neurons at stage 49 inherently exhibit lower intrinsic excitability, it would be difficult to observe a decrease in excitability. As such, we performed recordings on neurons at developmental stage 47/48, when intrinsic excitability of tectal neurons is higher (Pratt et al, 2008; Ciarleglio et al, 2015). As expected, the intrinsic excitability of control stage 47/48 tectal neurons was increased compared with stage 49 neurons (compare Fig 5E–G with Fig 1B). Crucially, we found that the intrinsic excitability of stage 47/48 tectal neurons exposed to MV1312 was significantly decreased compared with controls (Fig 5E and F), which shows that Na$^+$ currents carried by Na$_v$1.6 channels are important regulators of intrinsic excitability in *Xenopus* tectal neurons.

Tectal neurons, as the major recipient of sensory inputs in the optic tectum, play a crucial role in integrating multisensory information to produce an appropriate behavioral response (Deeg et al, 2009; Felch et al, 2016; Truszkowski et al, 2017; Busch & Khakhalin, 2019). As such, it is vital that tectal neurons can faithfully generate an action potential in response to graded sensory inputs. Therefore, we tested whether inhibition of Na$_v$1.6 channels with MV1312 alters the capacity of tectal neurons to repeatedly generate action potentials in response to a cosine-shaped injection of current. For these experiments, cells were exposed to repeated 200 ms cosine current injections at 30 Hz, with an increasing current amplitude from 40 to 120 pA (Fig 5G; top). The response frequency for each cosine current amplitude was calculated by measuring whether a spike was generated to each individual cosine wave. Significantly, whereas control cells could faithfully generate spikes at all current amplitudes, MV1312-exposed tectal neurons showed a lower response frequency at 40 pA (Fig 5G), which is consistent with Na$_v$1.6-mediated Na$^+$ currents being important for the response properties of tectal neurons. Taken together, these data suggest that Na$_v$1.6 channels contribute to all phases of the Na$^+$ currents and are important for determining tectal neuron action potential firing.

### Regulation of Na$^+$ channel subtype Na$_v$1.6 mediates homeostatic changes in Na$^+$ currents to control the intrinsic excitability of tectal neurons

We observed changes in the levels of expression of Na$_v$1.1 and Na$_v$1.6 channels in the optic tectum across tectal circuit development and in

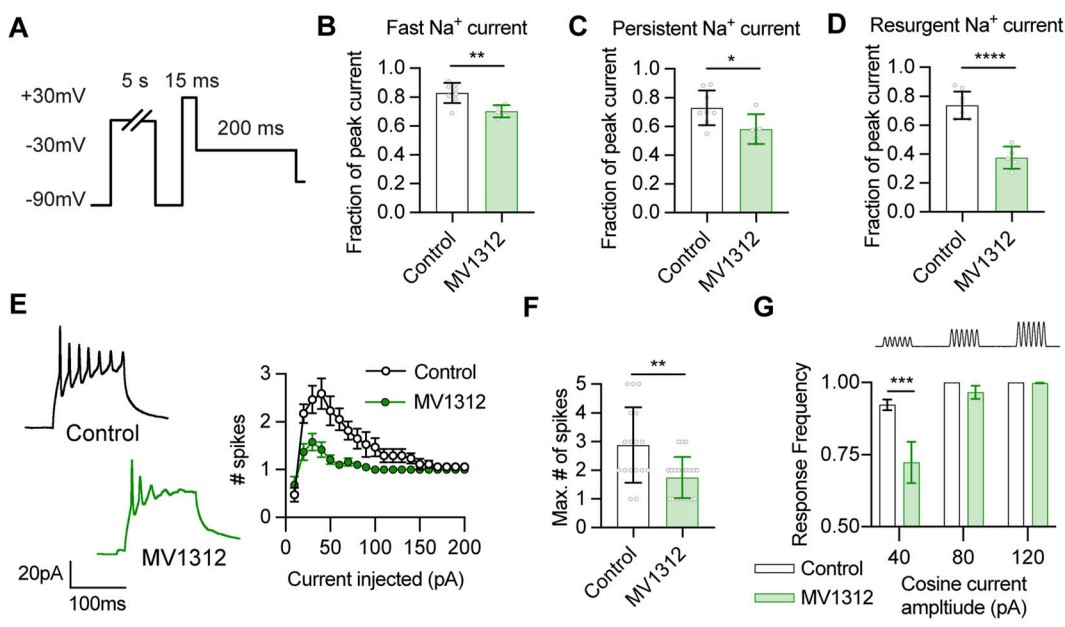

**Figure 5. The Na_v1.6 specific inhibitor MV1312 decreases intrinsic excitability by reducing fast, persistent, and resurgent Na⁺ currents in a use-dependent manner.**
**(A)** To measure the effect of Na_v1.6 channel inhibition on Na⁺ currents, we performed whole-cell recordings in the presence or absence of 5 µM of the specific Na_v1.6 channel inhibitor MV1312. We recorded Na⁺ currents before and after a 5 s depolarizing step to 0 mV to promote channel opening and drug binding, and then calculated the fraction of each Na⁺ current that remained. **(B, C, D)** Quantification of the fraction of initial peak current amplitude for the (B) fast Na⁺ current (Control: 0.82 ± 0.07, n = 8, MV1312: 0.70 ± 0.04, n = 5; $P$ = 0.0038), (C) persistent Na⁺ current (Control: 0.73 ± 0.12, MV1312: 0.58 ± 0.10; $P$ = 0.0439), and (D) the resurgent Na⁺ current (Control: 0.74 ± 0.10, MV1312: 0.38 ± 0.08; $P$ < 0.0001). Groups were compared with an unpaired $t$ test. **(E, F)** MV1312 decreases intrinsic excitability of tectal neurons by attenuating Na⁺ current amplitude. **(E, F, G)** The effect of MV1312 on intrinsic excitability was observed by measuring (E) spikes generated by current injection, (F) the maximum number of spikes generated by current injection (Control: 2.88 ± 1.32, n = 17; MV1312: 1.75 ± 0.72, n = 20; $P$ = 0.0021), and (G) the capacity of cells to spike in response to 200 ms cosine current injections at 30 Hz with increasing amplitudes from 40 to 120 pA (response frequency [% of current injections resulting in a spike]. 40 pA: Control 0.93 ± 0.02, n = 16, MV1312 0.72 ± 0.07, n = 19; $P$ = 0.0002; 80 pA: Control 1.00 ± 0.00, n = 16, MV1312 0.97 ± 0.02, n = 19; $P$ = 0.7429; 120 pA: Control 1.00 ± 0.00, n = 16, MV1312 0.99 ± 0.01, n = 19; $P$ = 0.9712. Groups were compared using a two-way ANOVA with a Holm–Sidak test for multiple comparisons).
Source data are available for this figure.

response to EVS in a similar manner to the regulation of Na⁺ currents and intrinsic excitability. We also found that inhibition of Na_v1.6 channels reduces intrinsic excitability. We therefore hypothesized that these Na⁺ channel subtypes may play an important role in the homeostatic regulation of intrinsic excitability. Knockout of Na_v1.1, Na_v1.2 or Na_v1.6 channels is lethal, resulting in prenatal death in rodent models (Catterall et al, 2010), whereas heterozygous loss of function of individual Na⁺ channel subtypes causes compensatory changes that often leads to neuronal and circuit hyperexcitability and seizures (Meisler et al, 2021). To determine the contribution of individual Na⁺ channel subtypes to homeostatic changes in Na⁺ currents and intrinsic excitability, we tested whether antisense morpholino RNA technology targeted to individual Na_v channel subtypes could suppress the EVS-mediated up-regulation of Na⁺ currents and intrinsic excitability. To achieve this, we bulk electroporated the tectum of stage 49 tadpoles with lissamine-tagged, translation-blocking antisense morpholino oligonucleotides (MO) specific for Na_v1.1, Na_v1.2 or Na_v1.6 channels, or a control MO. After 24 h, we performed whole-cell recordings from lissamine-positive tectal neurons and measured Na⁺ currents and intrinsic excitability from control tadpoles or tadpoles exposed to 4 h of EVS (Fig 6A).

We found that suppressing up-regulation of Na_v1.6 expression prevented the EVS-mediated increase in the fast, persistent, and resurgent Na⁺ currents; whereas suppressing up-regulation of Na_v1.1 expression prevented the EVS-mediated increase in the

persistent and resurgent Na⁺ currents, but did not prevent an increase in the fast Na⁺ current (Fig 6B–D and Table S3). In addition, we observed that knockdown of Na_v1.6 triggers a compensatory increase in the fast Na⁺ current compared with control MO (Fig 6B and Table S3), providing further evidence for a crucial role for Na_v1.6 channels in regulating excitability of tectal neurons. In contrast, we observed that up-regulation of Na_v1.2 expression was not required for EVS-mediated increase in the fast, persistent or resurgent Na⁺ currents (Fig 6B–D and Table S3). Taken together these data provide evidence that increased expression of Na_v1.6, and to a lesser extent Na_v1.1, is the molecular mechanism leading to for EVS-mediated increases in Na⁺ currents.

Our data indicated that the up-regulation of Na_v1.6 expression is required for EVS-mediated increases in Na⁺ currents, and that the specific Na_v1.6 channel blocker MV1312 decreases intrinsic excitability, suggesting that Na_v1.6 is a key regulator of Na⁺ currents that control homeostatic changes in intrinsic excitability. However, we had also observed that suppression of Na_v1.1 can reduce EVS-mediated increases in persistent and resurgent Na⁺ currents. Therefore, we next tested whether suppressing Na_v1.6 or Na_v1.1 expression attenuates the EVS-mediated homeostatic increase in intrinsic excitability. We found that suppressing up-regulation of Na_v1.6 expression prevented the EVS-mediated increases in intrinsic excitability, as shown by its effects on the input–output curves and the maximum number of spikes generated by current

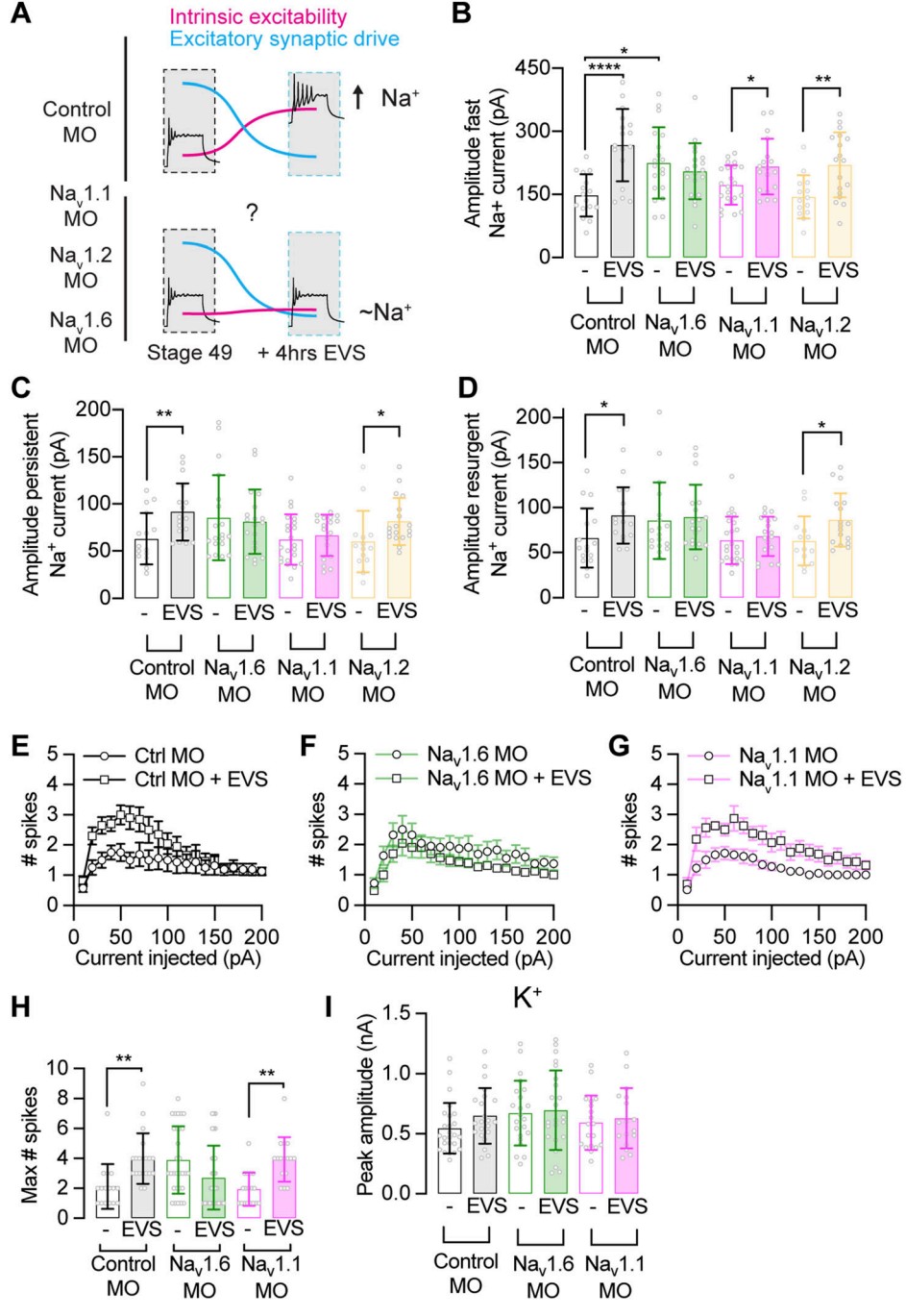

**Figure 6. Knockdown of Na_v1.6 attenuates network activity-dependent homeostatic increase in Na+ currents.**
**(A)** Schematic illustrates how exposure of stage 49 tadpoles to 4 h of enhanced visual stimulation (EVS) decreases excitatory synaptic drive and triggers a compensatory increase in intrinsic excitability via an increase in Na+ currents.
**(B, C, D)** Quantification showing the effect of acute knockdown of expression of specific Na_v channel subtypes on the EVS-triggered increases in the amplitude of the fast, persistent, and resurgent Na+ currents. Comparisons between experimental groups are presented in Table S3 (*n* = 16–22 cells). **(E, F, G, H)** Exposure of stage 49 tadpoles to 4 h of EVS triggers an increase intrinsic excitability, which is attenuated by the suppression of Na_v1.6 channel expression, but not Na_v1.1 channel expression (*n* = 16–31 cells). **(I)** K+ currents are not regulated with EVS-induced increases in intrinsic excitability. Comparisons between experimental groups are presented in Table S4. Groups were compared using a Welch's ANOVA test with Dunnett T3 test for multiple comparisons.
Source data are available for this figure.

step injection compared with control MO tadpoles (Fig 6E, F, and H and Table S4). In contrast, we found that inhibiting the up-regulation of Na_v1.1 expression has no effect on the EVS-mediated increases in intrinsic excitability (Fig 6E, G, and H and Table S4). As expected, there was no effect of EVS or Na_v1.6 MO on the peak amplitude of K+ currents (Fig 6I and Table S4), providing further evidence that regulation of Na+ currents is key to the control of intrinsic excitability in tectal neurons. Taken together, these data indicate that although EVS-induced increases in Na+ currents are mediated by both Na_v1.6 and Na_v1.1, but that it is Na_v1.6 up-

regulation that is critical for mediating the increase in excitability as measured by spike output.

## Perturbing expression of Na_v1.1 and Na_v1.6 channels during tectal circuit development causes deficits in sensorimotor behaviors

What are the functional consequences of Na+ channel subunit-mediated regulation of intrinsic excitability during development? The optic tectum, the primary visual area in the tadpole brain, undergoes activity-dependent refinement during development,

which is necessary for the accurate performance of visually guided behaviors (Dong & Aizenman, 2012; Khakhalin et al, 2014; Shen et al, 2014; James et al, 2015; Hamodi et al, 2016). Activity-dependent refinement of tectal circuitry from developmental stage 46 leads to better visual acuity as receptive field size decreases, and the temporal window for multisensory integration becomes narrower, whereas interventions that alter this activity-dependent process cause perturbed performance in tests of visual acuity, multisensory integration, and schooling behaviors (Schwartz et al, 2011; James et al, 2015; Truszkowski et al, 2016). One hypothesis is that the increase in excitability mediated by elevated $Na_v1.1$ and $Na_v1.6$ levels during developmental stage 46 is important for creating a permissive environment for activity-dependent plasticity required for proper tectal circuit development. Thus, we asked whether perturbing expression of $Na^+$ channel genes $Na_v1.1$ and $Na_v1.6$ during tectal circuit development causes behavioral deficits in tasks that require sensorimotor transformations such as visual acuity, multisensory integration, and schooling.

For these experiments, tadpoles at developmental stages 44–45 were co-electroporated with $Na_v1.1$ MO and $Na_v1.6$ MO, or with a control MO, to suppress $Na^+$ channel expression during a critical window of circuit development, with the behavioral tasks performed at stage 49. Importantly, electroporation of tadpoles with $Na_v$ channel-specific MOs or with $Na_v1.1$ MO + $Na_v1.6$ MO at stage 44–45 had no effect on the amplitude of the fast or persistent $Na^+$ currents at stage 49 (Table S5), which suggests that the effect of the channel expression knockdown on $Na^+$ current amplitude does not persist into the period of behavioral assessment. This is most likely because of the tectal neurons having compensated for the effect of the MOs on $Na^+$ channel expression (Van Wart & Matthews, 2006; Vega et al, 2008).

Visual acuity behavior correlates with the capacity of tectal neurons to tune visual spatial frequency sensitivity (Schwartz et al, 2011), providing a measure of tectal circuitry development. When presented with counterphasing gratings tadpoles change their swimming speed proportional to the spatial frequency of the gratings (Schwartz et al, 2011). We therefore examined whether perturbing expression of $Na^+$ channel genes $Na_v1.1$ and $Na_v1.6$ during tectal circuit development affects visual acuity behavior. Visual acuity responses were measured by calculating the change in velocity of tadpoles in response to the onset of a series of counterphasing sine wave gratings of different spatial frequencies presented (3, 4.5, 9, and 18 cycles/cm; Fig 7A). To analyze responses, we calculated a Z-score for individual tadpole to each spatial frequency, which describes how many standard deviations the mean response to each stimulus was from the mean response to no stimulus. To determine if tadpoles had responded to the visual stimuli, we tested whether the mean Z-score for control MO and $Na_v$ MO tadpoles to each spatial frequency was different from the no stimulus control. Control MO tadpoles showed responsiveness to gratings with low spatial frequency (Fig 6B), consistent with what has previously described (Schwartz et al, 2011). In contrast, $Na_v$ MO tadpoles showed decreased responsiveness to low spatial frequencies (Fig 7B), indicating a decreased visual acuity response. Importantly, this decreased the response in $Na_v$ MO tadpoles was not the result of decreased motility, as baseline motility was unchanged between control and $Na_v$ MO tadpoles (Fig 7C). These data

suggest that regulation of $Na_v1.1$ and $Na_v1.6$ expression levels during a critical period of tectal circuit development is necessary for the normal development of visual acuity response properties.

Multisensory integration (MSI), one of the primary functions of the tectum, is a highly conserved property of both neuronal output and behavior whereby the response to a stimulus of a single sensory modality is modulated by the coincident presentation of a stimulus from a different sensory modality. MSI depends on the strength of the neuronal response to each individual unimodal stimulus, the overlap between spatial receptive fields for the two sensory modalities, and the time window between presentations of the cross-modal pair (Wallace et al, 1998, 2006). As tectal neurons mature, they become more narrowly tuned to a more diverse range of interstimulus intervals, which occurs congruent with activity-dependent strengthening and refinement of synaptic connections (Felch et al, 2016). Because of its complex nature, MSI is a robust readout of connectivity deficits within the optic tectum. We therefore asked whether perturbing expression of $Na^+$ channel genes $Na_v1.1$ and $Na_v1.6$ during tectal circuit development affects MSI. Subthreshold visual or mechanosensory stimuli, or multisensory stimuli with interstimulus intervals of 0, 250, and 500 ms were presented to each tadpole (Fig 7D). Responses were determined by measuring normalized change in velocity of tadpoles to the stimulus onset (Fig 7E), from which we then calculated the multisensory (MS) index (Fig 7F). We observed that control tadpoles robustly respond to multisensory stimuli, with tadpoles increasing swimming to interstimulus intervals of 0 or 500 ms, and slowing down to 250 ms interstimulus intervals (Fig 7G). When we examined the preferred interstimulus intervals (ISI) of control tadpoles, we observed that tadpoles most strongly respond to multisensory stimuli with an ISI of 0 ms, with a few tadpoles preferring an ISI of 500 ms (Fig 7H). In contrast, $Na_v$ MO tadpoles exhibited a less robust response to multisensory stimuli and showed less temporal preference. $Na_v$ MO tadpoles exhibited a decreased MS index for both 0 and 500 ms (Fig 7F and G), with a shift in the preferred ISI that reflected how $Na_v$ MO tadpoles broadly responded to a wider range of ISI (Fig 7H), indicating a maturation of temporal tuning of multisensory responses.

Tadpoles perform a social aggregation behavior known as schooling, where tadpoles in close proximity engage in a coordinated unidirectional group swimming, which requires the integration of sensory cues (Wassersug & Hessler, 1971; Katz et al, 1981). Schooling tadpoles display short inter-tadpole distances and inter-tadpole angles less than 45° with neighboring tadpoles (depicted in Fig 7I), which is altered when tectal circuitry development is perturbed (James et al, 2015; Truszkowski et al, 2016). Because we had observed that perturbing $Na^+$ channel gene expression during tectal circuit maturation causes defects in visual acuity and multisensory integration behaviors, we predicted that perturbing expression of $Na_v1.1$ and $Na_v1.6$ during tectal circuit development would affect schooling behavior. Consistent with previous reports, the inter-tadpole distance of control tadpoles was ~2–3 cm, with control tadpoles most often found to be swimming in the same direction as their neighbors (Fig 7J and K). When we measured schooling behavior in $Na_v$ MO tadpoles, we observed that the inter-tadpole angles of $Na_v$ MO tadpoles was significantly altered, with fewer tadpoles swimming in the same direction (Fig 7K), but with no

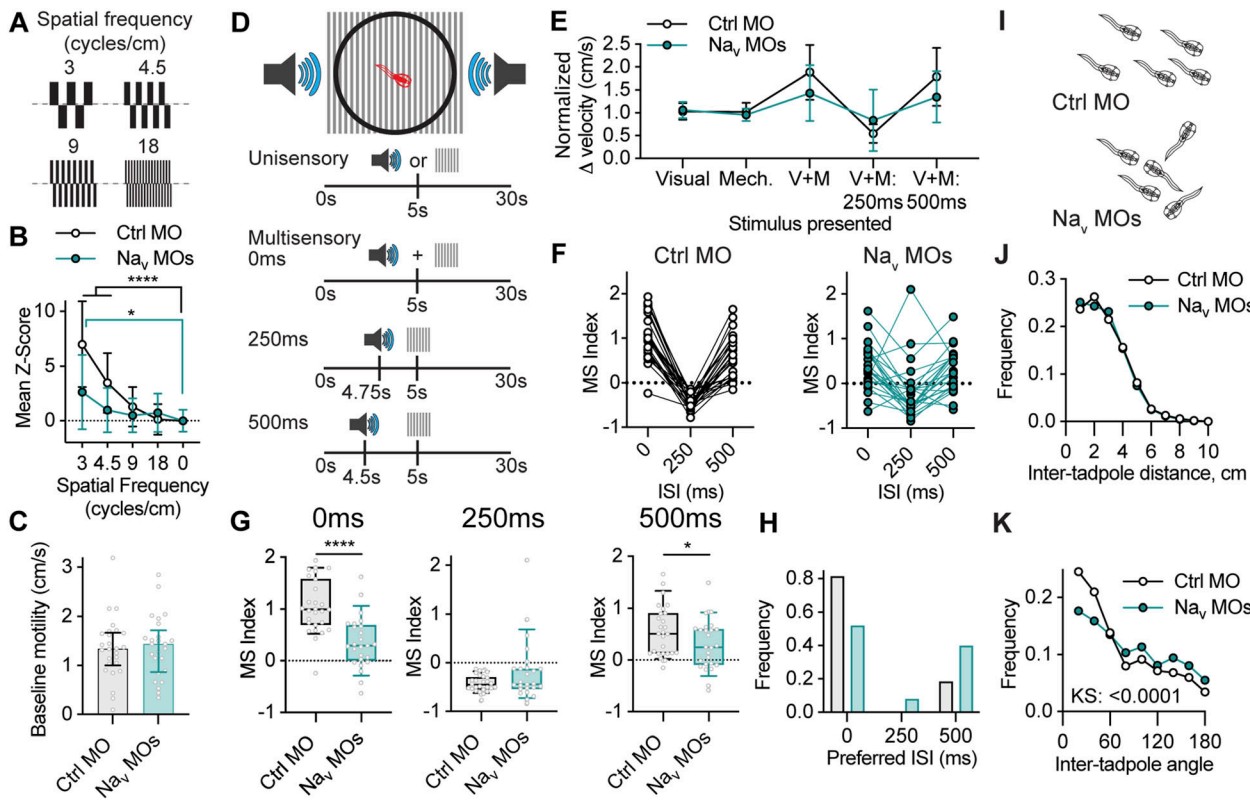

**Figure 7. Tadpoles in which expression of Na$_v$1.1 and 1.6 was perturbed during tectal circuit development show impairments in visual acuity, multisensory integration, and schooling behaviors.**

Effect of knocking down of Na$_v$1.1 and Na$_v$1.6 expression at stages 44–46 on tadpole behavior at stage 49 (comparisons showing that knockdown of Na$_v$1.1 and Na$_v$1.6 expression treatment had no effect on Na$^+$ currents at stage 49 are presented in Table S5). **(A, B, C)** Effect of developmental knockdown of Na$_v$ expression on visual acuity behavior. **(A)** Control morpholino (Ctrl MO) or Na$_v$1.1/Na$_v$1.6 morpholino (Na$_v$ MO) tadpoles were exposed to gratings counterphasing at 4 Hz over a range of spatial frequencies (3, 4.5, 9, and 18 cycles/cm). **(B)** Knockdown of Na$_v$1.1 and Na$_v$1.6 during the critical period of tectal development impaired responses to low spatial frequencies. Ctrl MO. 3 cycles/cm: 7.0 ± 3.9 ($P < 0.0001$), 4.5 cycles/cm: 3.5 ± 2.7 ($P < 0.0001$), 9 cycles/cm: 1.3 ± 1.8 ($P = 0.1238$), 18 cycles/cm: 0.1 ± 1.4 ($P = 0.8330$). Na$_v$ MO. 3 cycles/cm: 2.6 ± 3.4 ($P = 0.0107$), 4.5 cycles/cm: 1.0 ± 2.0 ($P = 0.9059$), 9 cycles/cm: 0.5 ± 1.6 ($P = 0.9056$), 18 cycles/cm: 0.7 ± 1.8 ($P = 0.5767$). $N = 7$ experiments of 3–4 animals. Groups were compared using a two-way ANOVA test with Holm–Sidak test for multiple comparisons. **(C)** This effect of Na$_v$ knockdown on visual acuity was not caused by a change in motility (Ctrl MO: 1.34, IQR = 1.00–1.67; Na$_v$ MO: 1.44, IQR = 0.87–1.72; $P = 0.8886$, $n = 25$–27 animals). Groups compared using a Mann–Whiney $U$ test. **(D, E, F, G, H)** Effect of developmental knockdown of Na$_v$ expression on multisensory integration behavior. **(D)** Experimental paradigm illustrating the presentation of visual and mechanosensory stimuli, or multisensory stimuli with interstimulus intervals ranging from 0–500 ms. **(E)** Mean normalized change in velocity (cm/s) in response to unisensory visual or mechanosensory stimuli, or multisensory stimuli, for Control MO and Na$_v$ MO tadpoles. **(F)** Multisensory (MS) indexes calculated for individual tadpoles at 0, 250, and 500 ms. Values for individual tadpoles are connected by a line. **(G)** Quantification of MS index for interstimulus intervals of 0 ms (Ctrl MO: 1.1 ± 0.5, Na$_v$ MO: 0.4 ± 0.5, $P < 0.0001$), 250 ms (Ctrl MO: −0.4 ± 0.2, Na$_v$ MO: −0.2 ± 0.6, $P = 0.0713$) and 500 ms (Ctrl MO: 0.6 ± 0.5, Na$_v$ MO: 0.3 ± 0.4, $P = 0.0438$). Groups were compared using Welch's $t$ test. **(H)** Histogram of preferred interstimulus intervals for each tadpole. $N = 8$ experiments of 3–4 animals. **(I, J, K)** Effect of developmental knockdown of Na$_v$ expression on schooling behaviour. **(I)** Schematic illustrates aggregated schooling behavior observed in control MO and Na$_v$ MO tadpoles, with Na$_v$ MO tadpoles observed to be less likely to be swimming in the same direction, without a change in inter-tadpole distance. **(J, K)** These observations are quantified by observing (J) inter-tadpole distance (Ctrl MO: 2.4 cm, IQR = 1.4–3.6 cm; Na$_v$ MO: 2.4 cm, IQR = 1.3–3.6 cm; $P = 0.4088$), and (K) inter-tadpole angles (Ctrl MO: 45.7°, IQR = 19.7–93.9°; Na$_v$ MO: 65.6°, IQR = 29.2–119.1°; $P < 0.0001$). $N = 6$ experiments of 20 animals per experimental group. Groups were compared using a Kolmogorov–Smirnov test.
Source data are available for this figure.

effect on the inter-tadpole distance (Fig 7J and illustrated in Fig 7I). These data show how perturbing the regulation of intrinsic excitability by impairing regulation of Na$^+$ channel gene expression during circuit development alters circuit function and causes impaired schooling behavior, most likely by perturbing multisensory integration in the tectum.

All these behavior tasks require proper activity-dependent refinement of tectal circuits. We observed that tadpoles in which the expression of Na$_v$1.1 and Na$_v$1.6 was perturbed during tectal circuit development shows reduced visual acuity responsiveness, broader multisensory tuning, and abnormal schooling behavior. These data

illustrate the vital importance of the correct regulation of neuronal excitability by Na$^+$ channel gene expression for normal tectal circuit development.

## Discussion

In this study, we explored the molecular mechanisms by which neurons of the optic tectum adapt their intrinsic excitability during circuit development and with sensory experience. We demonstrate

that this process primarily relies on the dynamic adaption of Na$^+$ currents occurring via the regulation of Na$_v$1.6, and to a lesser extent Na$_v$1.1, channel expression. We then show the importance of this mechanism of regulating excitability for normal circuit development by demonstrating that dysregulation of Na$_v$1.6 and Na$_v$1.1 channel expression during a critical phase of tectal circuit development, when the circuitry is undergoing activity-dependent refinement, causes deficits in behaviors reliant on visual and multisensory processing. Our findings illustrate the vital importance that the regulation of Na$_v$ channel expression plays in controlling Na$^+$ currents for the homeostatic control of neuronal excitability in the developing nervous system, and for the first time show that regulation of Na$^+$ channel expression is an important mechanism for intrinsic plasticity in the midbrain.

### A key role for persistent and resurgent Na$^+$ currents in regulating homeostatic changes in neuronal excitability

In our search for the cellular mechanisms that regulate homeostatic changes in excitability during retinotectal circuit development, we identified distinct persistent and resurgent Na$^+$ currents in neurons of the optic tectum. Persistent currents had previously been observed in these neurons (Aizenman et al, 2003; Hamodi & Pratt, 2014), but they had not been characterized, nor had their function been explored. Whereas this is the first report that *Xenopus* tectal neurons express a resurgent Na$^+$ current, these Na$^+$ currents broadly resemble those described in mammalian neurons, but with some important differences. In mammalian neurons, persistent and resurgent Na$^+$ currents usually represent a small fraction of the fast Na$^+$ current (~1 and ~5–10%, respectively [Lewis & Raman, 2011]). In contrast, *Xenopus* neurons have relatively large persistent and resurgent Na$^+$ currents. Across all of the developmental stages studied, the amplitude of the persistent Na$^+$ current was ~40% of fast Na$^+$ current, and the amplitude of the resurgent Na$^+$ current was ~35–50%. One interpretation of these data is that fast currents have yet to have fully developed in these relatively immature neurons. Because tectal neurons generally have relatively small fast Na$^+$ currents (~300 pA) compared with the nA-sized currents observed in mammalian neurons, it is possible that as tectal neurons continue to mature their fast Na$^+$ current will increase relative to persistent and resurgent Na$^+$ currents. Another interpretation is that these relatively large persistent and resurgent Na$^+$ currents allow tectal neurons to rapidly adapt the action potential waveform and, therefore, their excitability. Propagation of postsynaptic potentials to the soma causes graded voltage change that, if larger than a certain threshold, leads to the initiation of an action potential. Hence, the integrative function of a neuron is strongly affected by changes in depolarizing currents, including persistent and resurgent Na$^+$ currents, that drive membrane voltage closer to threshold potential (Connors & Prince, 1982; Stafstrom et al, 1984; Raman et al, 1997). As tectal neurons develop and exist in an environment where sensory experience is continuing to alter synaptic organization and strength, it is conceivable that they require mechanisms to rapidly adapt to high and low levels of synaptic input, which could be achieved via the regulation of persistent and resurgent Na$^+$ currents. Our observation that persistent and resurgent Na$^+$ currents are regulated with changes in

excitability across development and in response to patterned visual experience supports this idea. Furthermore, our confirmation that K$^+$ currents are not regulated with changes in excitability during development or in response to patterned visual experience does not rule out a role for K$^+$ currents in the regulation of excitability; however, our findings are consistent with regulation of Na$^+$ currents being the key mechanisms used by tectal neurons to adapt their intrinsic excitability during normal circuit development.

The structure of tectal neurons and their patterns of projection mean that the axon initial segment (AIS) is located within the dendritic arbor (Haas et al, 2006; Chiu et al, 2008; Marshak et al, 2012), which may impact the measurement of Na$^+$ currents using whole-cell recordings at the soma (Spruston et al, 1993). We know that Na$^+$ currents, including persistent and resurgent currents, in subcellular compartments such as the AIS have a relatively large influence on neuronal excitability (Kole et al, 2008; Hu et al, 2009; Osorio et al, 2010; Gorski et al, 2018; Hsu et al, 2018; Shvartsman et al, 2021). As discussed, tectal neurons are small with correspondingly short dendrites and relatively small and slow fast Na$^+$ currents. Because measurements were restricted to peak current amplitude, and not activation and inactivation kinetics because of possible space-clamp issues as discussed previously (Aizenman et al, 2003; Pratt & Aizenman, 2007), we believe that changes in fast, persistent, and resurgent currents observed across development and with EVS reflect changes in Na$^+$ channel function and not a change in our ability to resolve Na$^+$ currents.

### Na$_v$1.6 is a molecular mediator of neuronal excitability in the developing *Xenopus* visual system

Tectal neurons homeostatically regulate the amplitude of their fast Na$^+$ currents in responses to changes in synaptic input caused by experience-dependent developmental changes in circuit architecture (Pratt & Aizenman, 2007; Ciarleglio et al, 2015), short-term changes in visual experience (Aizenman & Linden, 2000; Ciarleglio et al, 2015), and in response to the overexpression of K$^+$ channels (Pratt & Aizenman, 2007; Dong & Aizenman, 2012). In this study, we extend these findings by describing how dynamic changes in Na$_v$1.6 channel expression functions as a molecular mechanism to adapt Na$^+$ currents in response to changing synaptic inputs, and how this modulation of Na$^+$ current amplitudes is required to mediate homeostatic changes in excitability.

Using ion substitution experiments, pharmacology, and RNA interference technology, we determined that *Xenopus* tectal neurons express TTX-insensitive persistent and resurgent Na$^+$ currents that are largely mediated by Na$_v$1.6 channels. This was initially puzzling because *Xenopus* do not express traditional neuronal TTX-resistant Na$^+$ channels Na$_v$1.8 or Na$_v$1.9. Nevertheless, whereas TTX abolished the fast Na$^+$ current, we observed no effect of TTX on the persistent and resurgent Na$^+$ currents.

Evidence from electrophysiology experiments performed in other anuran species and analysis of the protein structure of *Xenopus* Na$_v$1.6 provides support for our observation of TTX-resistant Na$^+$ currents in *Xenopus* tectal cells. TTX-resistant Na$^+$ currents have been reported in neurons of other frog species (Campbell, 1992a, 1992b; Kobayashi et al, 1993, 1996). Neurons isolated from the common frog express both TTX-sensitive and TTX-

resistant (up to 100 $\mu$M TTX) Na$_v$ channels, with TTX-resistant channels inactivating two to six times more slowly than TTX-sensitive channels (Campbell, 1992a). In addition, neurons isolated from the bullfrog generate a Na$^+$-dependent spike that is resistant to TTX (Kobayashi et al, 1993, 1996). This observed resistance to TTX in *Xenopus* is perhaps less surprising because anuran species express fewer Na$_v$ subtypes (Zakon et al, 2011; Zakon, 2012), and many species have cutaneous secretions that include TTX (Mebs & Schmidt, 1989; Yasumoto & Yotsu-Yamashita, 1996; Pires et al, 2002, 2005). Indeed, the TTX-resistant Na$_v$1.6 channel generated by mutating tyrosine to serine within the P-loop of domain I to prevent TTX binding in the channel pore (Herzog et al, 2003) targets the very same tyrosine that is substituted by phenylalanine in *X. laevis* homologs of Na$_v$1.6. A substitution that is also present in the closely related species *Xenopus tropicalis* and the common toad (*Bufo bufo*). The presence of this substitution in *Xenopus* may explain our observation of TTX-resistant persistent and resurgent Na$^+$ currents. Determining how the tyrosine to phenylalanine substitution underlies this distinct effect of TTX on Na$_v$1.6 currents in *Xenopus* tectal neurons is an interesting question meriting further exploration, but is outside the scope of this study.

Of the Na$^+$ channel subtypes expressed in the *Xenopus* brain (Na$_v$1.1, Na$_v$1.2, and Na$_v$1.6), Na$_v$1.6 have been strongly linked with regulation of persistent and resurgent Na$^+$ currents (Raman et al, 1997; Khaliq et al, 2003; Enomoto et al, 2007; Patel et al, 2015). We therefore hypothesized that Na$_v$1.6 channels may have a role in regulating persistent and resurgent Na$^+$ currents and, hence, homeostatic changes neuronal excitability. This hypothesis was supported by our findings that the specific Na$_v$1.6 channel inhibitor MV1312 attenuated Na$^+$ currents, and that antisense RNA technology targeted to Na$_v$1.6 prevented the visual experience-dependent up-regulation of fast, persistent, and resurgent Na$^+$ currents; thus, preventing a homeostatic increase in intrinsic excitability. In contrast, suppression of Na$_v$1.1 expression did not attenuate the visual experience-dependent up-regulation of the fast Na$^+$ current or intrinsic excitability. When considered together with our observation that Na$_v$1.1 expression increases across development and not with changes in excitability during development, whereas only being subtly increased compared with the large up-regulation of Na$_v$1.6 expression after visual experience, this suggests that there is a dissociation in the regulation of different Na$_v$ channel subunits to regulate the intrinsic excitability of tectal neurons. However, a specific Na$_v$1.1 inhibitor would be required to fully characterize the contribution of these channels to the regulation of Na$^+$ currents and excitability in tectal neurons. Nevertheless, these findings have important implications for our understanding of the molecular mechanisms that regulate homeostatic changes in neuronal intrinsic excitability for experience-dependent development of the retinotectal circuitry.

### How are Na$^+$ channels regulated to mediate homeostatic changes in intrinsic excitability?

What are the mechanisms by which Na$^+$ channels are modulated to regulate homeostatic changes in Na$^+$ currents and intrinsic excitability? The amplitude and kinetics of Na$^+$ currents are determined by the specific repertoire of Na$^+$ channel subtypes expressed, by the

compartmentalized localization and density of channels, and by the regulation of channel kinetics (Bean, 2007; Leterrier et al, 2011; Wood & Iseppon, 2018; Sole & Tamkun, 2020). Here, we identified changes in Na$^+$ channel gene expression that occur together with changes in both Na$^+$ current amplitude and intrinsic excitability across development and with experience-dependent plasticity, albeit with some differences. Notably, expression of Na$_v$1.1 increased with development, whereas the peak of Na$_v$1.6 expression corresponded with the peak of neuronal excitability during tectal circuit development. Whereas 4 h exposure to visual stimulation caused a large increase in expression of Na$_v$1.6 and Na$_v$4$\beta$, together with a smaller increase in Na$_v$1.1 channel expression. Although these findings support the idea that regulation of Na$_v$1.6 channel gene expression is a key mechanism by which Na$^+$ currents are modulated to mediate experience-dependent changes in excitability of tectal neurons, they also suggest that there are differences in the mechanisms underlying the modulation of Na$^+$ currents and intrinsic excitability in these two conditions.

Are different mechanisms at play as neurons adapt their intrinsic excitability with developmental experience-dependent changes in synaptic input versus the short-term homeostatic increase in excitability resulting from enhanced sensory experience decreased synaptic input? Although we observed a strong correlation between developmental changes in excitability and Na$^+$ currents, the correlation between Na$_v$1.6 channel gene expression and these two biophysical properties was less apparent when comparing developmental stages 46 and 49. One interpretation for these results is that exposing tadpoles to EVS causes coordinated and widespread suppression of synaptic inputs from retinal ganglion cells across the tectum leading to a synchronized increases in excitability of tectal neurons within hours (Aizenman et al, 2003; Ciarleglio et al, 2015), as opposed to a more subtle change in synaptic inputs occurring over several days (Aizenman & Cline, 2007; Ciarleglio et al, 2015). These differences in timescales can explain the differences observed in the regulation of expression of Na$^+$ channel subtypes with developmental changes in excitability compared with short-term experienced dependent changes. However, these data do not rule out the possibility that mature and immature neurons use different mechanisms to regulate Na$^+$ channel function and, therefore, their intrinsic excitability.

One mechanism that could be specific to mature tectal neurons is the regulation of Na$^+$ channel function by accessory proteins such as Na$_v$4$\beta$. Na$_v$4$\beta$ becomes enriched at the AIS in a manner that depends on Na$^+$ channel localization (Buffington & Rasband, 2013), has been found to increase in expression with maturation of neurons (Zemel et al, 2021), and has a well-defined role modulating resurgent Na$^+$ currents (Grieco & Raman, 2004; Bant & Raman, 2010). Na$_v$4$\beta$ is well-placed to contribute to the regulation of Na$^+$ currents as neurons adapt their firing rate in response to short-term changes in synaptic input. Given that we found a large increase in Na$_v$4$\beta$ expression after EVS, with little evidence for any developmental regulation, future studies could explore the contribution of Na$_v$4$\beta$ to changes in Na$^+$ current kinetics and whether regulation of Na$_v$4$\beta$ expression is a mechanism used by mature tectal neurons to control dynamic changes in their intrinsic excitability.

Na$^+$ channel subtypes are regulated both in a cell-specific manner and with developmental patterns by both alternative splicing and

translational repression/activation mechanisms. In the mammalian CNS, the splicing pattern of Na$^+$ channel subtypes (Na$_v$1.1–3 and Na$_v$1.6) varies across development, with mutually exclusive protein-coding exons found in neonates versus adults (Gustafson et al, 1993; Kasai et al, 2001; Copley, 2004; O'Brien et al, 2012; Liang et al, 2021). Alternative splicing and translational control of Na$^+$ channel subtypes are also performed in an activity-dependent manner. In flies and rodents, activity-dependent regulation of Na$^+$ channel gene expression involves the translational regulator Pumilio (Mee et al, 2004; Lin et al, 2012; Driscoll et al, 2013; Lin & Baines, 2015). Activity-dependent splicing of the *Drosophila* Na$^+$ channel modulates the persistent Na$^+$ current to regulate excitability (Muraro et al, 2008; Lin et al, 2009). It is not known whether similar mechanisms to control Na$^+$ channel gene expression is present in tectal neurons.

What is the functional importance of the increase in Na$^+$ channel gene expression observed in response to enhanced sensory experience? We observed increases in Na$^+$ channel gene expression that correlated with both increased Na$^+$ current amplitude and increased intrinsic excitability after 4 h of EVS, which raises questions about how Na$^+$ channels are being regulated, including whether channels are being translated, folded, and inserted into the membrane during this time period. Whereas studies conducted in vitro suggest that the half-life of Na$^+$ channels is between 24–35 h (Ritchie, 1988; Hildebrand et al, 1993; Maltsev et al, 2008), imaging of single channels in neurons would suggest that Na$^+$ channel availability at the plasma membrane is a much more dynamic process. In rodent hippocampal neurons, fluorescently tagged Na$_v$1.6 channels are directly inserted into the soma and AIS from local endocytic vesicles, with ~10% of Na$_v$1.6 channels located at the AIS being inserted within 25 min (Akin et al, 2015; Gonzalez et al, 2016). This would fit with Na$^+$ channels becoming localized to the plasma membrane in response to posttranslation modifications, including glycosylation (Waechter et al, 1983; Schmidt et al, 1985; Schmidt & Catterall, 1986), allowing for an increase in channel surface density faster than de novo synthesis would allow in response to mediate the homeostatic increases in excitability resulting from visual stimulation-induced synaptic scaling. Rapid removal of Na$^+$ channels from the plasma membrane can also occur. CA1 pyramidal neurons exposed to NMDA receptor agonists undergo activity-dependent plasticity of the AIS that involves clathrin-mediated endocytosis of Na$^+$ channels (Fréal et al, 2023). In these experiments, the AIS length is decreased 30 min after NMDA application, correlating with changes in the distribution of Na$^+$ channels at the AIS. These studies illustrate the dynamic regulation of Na$^+$ channel availability. Although these studies suggest that the initial phase of the homeostatic increase in excitability observed after EVS may be independent of translation, they do not rule out a role for translation of new Na$^+$ channels. Indeed, studies conducted in *Xenopus* tadpoles provide evidence that changes in protein expression are required for structural and functional plasticity of tectal cells and circuitry in response to short-term sensory experience. It has previously been reported that 30 min of visual conditioning is sufficient to cause plasticity of visual avoidance behaviors, an effect that requires the synthesis of new proteins (Shen et al, 2014; Liu & Cline, 2016; Liu et al, 2018), and degradation of proteins via the neuronal membrane proteasome (He et al, 2023). Together, these studies illustrate the important role of protein

homeostasis as neurons functionally adapt in response to changing sensory input.

When these studies are considered together with our observation of a large increase in Na$^+$ channel expression in response to EVS, they provide evidence for a two-step model whereby a readably available pool of Na$^+$ channels become inserted into the plasma membrane, whereas newly expressed channels contribute to both replacing this intracellular pool and increasing channel density on the membrane long-term. Crucially, support for this model is provided by our results showing that the increases in Na$^+$ currents and excitability induced by 4 h exposure to EVS is attenuated by Na$_v$1.6 MO. While this model would fit with a recent study in which tadpoles exposed for 4 h to sensory cues with lower saliency found that tectal neurons do not display an increase in intrinsic excitability, which was thought to result from a lack of synaptic downscaling in response to these exposures (Busch & Khakhalin, 2019). Furthermore, it is also noteworthy that mRNA for voltage-gated Na$^+$ channels has previously been detected in the axonal compartment of *Xenopus* neurons (Zivraj et al, 2010), meaning that it is conceivable that local synthesis of Na$^+$ channels may contribute to the homeostatic plasticity of Na$^+$ currents and excitability, especially for Na$^+$ channels in the axonal or dendritic compartments where increased persistent and resurgent currents could be important for the amplification of synaptic inputs or AP propagation. How the synthesis, trafficking, and degradation of Na$^+$ channel subtypes, including any role for local translation, occur in tectal during homeostatic plasticity is an interesting question that warrants further study.

If increased Na$^+$ channel expression can mediate homeostatic increases in excitability, what mechanisms mediated the homeostatic decrease in excitability? We have discussed the importance of clathrin-mediated endocytosis as a mechanism that can actively retrieve Na$^+$ channels from the plasma membrane in response to changes in synaptic function (Fréal et al, 2023), and the importance of activity-dependent transcriptional control mechanisms, such as the transcriptional repressor Pumillo in regulating Na$^+$ channel expression (Mee et al, 2004; Lin et al, 2012; Driscoll et al, 2013; Lin & Baines, 2015), and protein degradation by mechanisms such as the neuronal membrane proteasome (He et al, 2023). Phosphorylation is also an important modulator of Na$^+$ channel function. Phosphorylation of Na$^+$ channels has long been associated with reduced currents and slowing of channel inactivation (Numann et al, 1991; Carr et al, 2003). Phosphorylation can also regulate Na$^+$ channel expression and localization. For example, the interaction between Na$^+$ channels and the scaffolding protein ankyrin G is regulated by phosphorylation of several serine residues by CK2 (Bréchet et al, 2008; Montenarh & Gotz, 2020). Here, we observed a trend towards decreased expression of Na$_v$1.6 expression between developmental stages 46 and 49, correlating with the decrease in Na$^+$ currents between these stages, which suggests that reducing the expression of Na$_v$1.6 channels contributes to the decrease in excitability between these stages. However, the mechanisms that are responsible for this decrease in Na$_v$1.6 channel expression remain to be determined.

To better understand the mechanism by which changes in synaptic input are sensed and translated into a change in Na$^+$ current amplitude, and whether this involves the regulation of

individual Na$^+$ channel subtypes or all Na$^+$ channel subtypes, future experiments should explore the molecular mechanisms by which Na$^+$ channel gene expression is regulated as tectal neurons undergo homeostatic plasticity of intrinsic excitability. Deciphering the cellular mechanisms by which tectal neurons regulate Na$^+$ channel expression and function during retinotectal circuit development will be crucial for understanding how neurons can be flexible and respond to massive changes in circuit and synaptic organization with development or in response to sensory experience while maintaining the ability to robustly respond to synaptic inputs.

### Functional implications of subtype-specific homeostatic plasticity of Na$^+$ currents

Early life sensory experience during critical periods of development shapes circuits in the brain (Fox, 1992; Crair & Malenka, 1995; Hensch & Fagiolini, 2005). These critical periods represent time windows when neurons and circuits are particularly sensitive to modification. It is thought that disruption of activity-dependent homeostasis processes during these critical periods contributes to neurodevelopmental disease such as autism spectrum disorder and epilepsy (Meredith et al, 2012; Doll & Broadie, 2014; Mullins et al, 2016). Given their role in action potential generation, Na$_v$1.6 channels, are well placed to be key regulators of neuronal excitability. Indeed, both gain-of-function and loss-of-function mutations in Na$^+$ channel genes can trigger neuronal and circuit dysfunction that results in neurodevelopmental disorders including epilepsy and autism spectrum disorder (Cannon & Bean, 2010; Meisler et al, 2021). In the mammalian brain, Na$_v$1.2 is replaced by Na$_v$1.6 as the major constituent of the AIS during postnatal development (Caldwell et al, 2000; Boiko et al, 2001). Neurons of Na$_v$1.6-deficient mice found to be have reduced resurgent currents and impaired ability to fire repetitively (Raman et al, 1997; Khaliq et al, 2003; Van Wart & Matthews, 2006), even though there is a compensatory increase in Na$_v$1.2 at the AIS and nodes of Ranvier in these Na$_v$1.6-deficient neurons (Vega et al, 2008). These findings suggest that there may be a subtype specific role of Na$_v$1.6 in regulating neuronal intrinsic excitability, consistent with our observations in tectal neurons.

One possibility is that Na$_v$1.1 and Na$_v$1.6 channels have distinct functions in developing tectal neurons conferred by their intracellular localization, conductance properties, or association with accessory proteins such as Na$_v$4$\beta$, in similar manner to that described for mammalian neurons (Lewis & Raman, 2011; Theile & Cummins, 2011; Buffington & Rasband, 2013; Sun et al, 2015; Ransdell et al, 2017). Unfortunately, we do not currently have the tools required to assess the subcellular localization of Na$_v$1.1 and Na$_v$1.6 channels and any functional relevance that this might have for tectal neurons.

However, consistent with this idea, we observed that Na$_v$1.6 is transiently up-regulated at developmental stage 46, which reflects a key period of tectal circuit maturation where activity-dependent refinement and strengthening of synaptic connections. We also found that Na$_v$1.6 and the accessory subunit Na$_v$4$\beta$, which has been shown to regulate resurgent Na$^+$ currents by binding to Na$_v$1.6 channels, were both highly up-regulated after exposure to sensory experience. Furthermore, when we prevented up-regulation of Na$_v$1.6 and Na$_v$1.1 expression during the critical period of tectal circuit development at stage 46, we observed deficits in behaviors that rely on sensory integration in the tectum. These findings suggest that perturbing Na$^+$ channel mediated regulation of excitability is sufficient to cause deficits in circuit function. By showing the Na$^+$ channel subtype-specific role in regulating excitability, our work also illustrates the utility of the *Xenopus* visual system as an experimental model for studying the effect of Na$^+$ channelopathies on neuronal and circuit development.

# Materials and Methods

### Animals

All animal experiments were performed in accordance with and approved by Brown University Institutional Animal Care and Use Committee standards. *X. laevis* tadpoles were raised in 10% Steinberg's solution on a 12 h light/dark cycle at 18–21°C, with developmental stages determined according to established standards (Nieuwkoop & Faber, 1994). Under these rearing conditions, tadpoles generally reach stage 42 at 6–7 days postfertilization (dpf), stage 46 at 9–12 dpf, and stage 49 at 18–20 dpf.

### EVS

To trigger homeostatic increases in the intrinsic excitability of tectal neurons, tadpoles were exposed to short-term (4 h) EVS at developmental stage 49. Freely swimming tadpoles were transferred to a custom-built light chamber consisting of four rows of three LEDs that flashed in sequence at 1 Hz to simulate motion stimuli (Sin et al, 2002; Aizenman et al, 2003; Ciarleglio et al, 2015). Brains were prepared for electrophysiology or RNA extraction immediately after EVS exposure.

### Morpholinos

Knockdown of expression of specific Na$^+$ channel subtypes was achieved using 3′-lissamine-tagged translation-blocking antisense morpholino oligonucleotides (MO; GeneTools) targeted to *Xenopus* Na$_v$1.1 (5′-TTACTGCTTTGCTACTTTCATAATG-3′), Na$_v$1.2 (5′-GTGGTTGCTCCATCTTCTCATCC-3′), and Na$_v$1.6 (5′-CAACTTCTCCTGTTAAG-TAGCGCCT-3′). To control off-target effects of MO electroporation, we used a 3′-lissamine-tagged Control MO (5′-CCTCTTACCTCAGTTA-CAATTTATA-3′). MOs were dissolved in water at 0.1 mM. To electroporate MOs into cells of the optic tectum, MOs were injected into the brain ventricle before platinum electrodes were placed on either side of the midbrain and three to five pulses at ~30 V with an exponential decay of 70 ms applied bidirectionally, which results in bulk electroporation of tectal neurons.

### Quantification of Na$^+$ channel subtype expression levels

To measure Na$^+$ channel subtype expression levels during development, whole tecta were harvested from tadpoles at developmental

stages 42, 46 or 49. To determine how $Na^+$ channel subtype expression was affected by exposure to EVS, whole tecta were harvested from stage 49 tadpoles exposed to 4 h EVS or untreated stage 49 control tadpoles. For each experimental group, optic tecta were collected from 10 tadpoles from three independent breedings and stored in RNAlater before the isolation of RNA by TRIzol extraction. SuperScript IV VILO first strand cDNA synthesis was then performed according to the manufacturer's protocol (Invitrogen). Real-time quantitative PCR (RT–qPCR) was performed by using Power SYBR Green master kit with an Applied Biosystems StepOnePlus according to standard manufacturer's protocols (Thermo Fisher Scientific). Data were analyzed by the ΔΔCT relative quantification method using the housekeeper gene RSP13 and are represented as a fold change in expression from a control condition (developmental stage 42 or naïve stage 49 control). Primer sequences (forward and reverse) for SCN1A (5′-GCAATGGCCACCAACTGAC-3′ and 5′-AATCAGAGGAGTTACCACAGAGC-3′), SCN2A (5′-GCTGGCTTAAAA-TAAAGCATGTACT-3′ and 5′-TAGCTTGAAAACCATCTCGGCA-3′), SCN8A (5′-TGTGTGGCGCGTTTTAAGATTT-3′ and 5′-TTCGAATGGTTTTCCGCTGTTC-3′), and SCN4B (5′-GCAAGAATAACCTGGTCACAGC-3′ and 5′-CAGGTTTAGG-GAATGACTTCTTGTC-3′). The RSP13 primer sequences have previously been published (Thompson & Cline, 2016).

## Electrophysiology

For whole-cell recordings from tectal neurons tadpole brains were prepared as described previously to access the ventral surface of the tectum (Wu et al, 1996). In brief, tadpoles were anaesthetized in 0.02% tricaine methanesulfonate (MS-222), before brains were filleted along the dorsal midline, removed, and pinned to a sylgard block submerged in a recording chamber and maintained at RT for the duration of the experiment. Unless otherwise stated, brains were maintained in HEPES-buffered extracellular saline (in mM: 115 NaCl, 2 KCl, 3 CaCl$_2$, 3 MgCl$_2$, 5 HEPES, 10 glucose, pH 7.2, Osmolarity: 250 mOsm), for the duration of the experiment (typically 2–3 h). To access principal tectal neurons, the ventricular membrane was removed by suction using a broken glass pipette. Recordings were restricted to the middle one-third of the tectum to avoid any developmental variability along the rostrocaudal axis (Khakhalin & Aizenman, 2012; Hamodi & Pratt, 2014). Cells were visualized using a Nikon eclipse E600FN light microscope equipped with a 60x water-immersion objective, a Lumencor SOLA light engine for fluorescent illumination, and a Hamamatsu IR-CCD camera. To record from MO electroporated neurons, lissamine-tagged MOs were visualized using fluorescence before switching to an IR-filter for recordings. Whole-cell voltage-clamp recordings were performed using glass micropipettes (10–12 MΩ). Electrical signals were measured with a Multiclamp 700B amplifier, digitized at 10 kHz using a Digidata 7550B low-noise analog-to-digital board, and acquired using pClamp 11 software (Molecular Devices). Active currents were isolated by leak subtraction performed in real time using pClamp software. Membrane potential was not adjusted for a predicted 12 mV liquid junction potential. Neurons with series resistance >50 MΩ were not included in the data set. Recordings were analyzed using Axograph X software (John Clements).

To record mixed-current responses, pipettes were filled with K-gluconate intracellular saline solution (in mM: 100 K-gluconate, 8 KCl, 5 NaCl$_2$, 1.5 MgCl$_2$, 20 HEPES, 10 EGTA, 2 ATP disodium salt hydrate, 0.3 GTP sodium salt hydrate, pH 7.2, Osmolarity: 255 mOsm). To record $Na^+$ currents in the absence of $K^+$ currents, pipettes were filled with Tris-based intracellular saline solution (in mM: 67 Tris–PO$_4$, 73 Tris–OH, 20 TEA-Cl, 10 EGTA, 10 sucrose, 2 ATP disodium salt hydrate, 0.1 GTP sodium salt hydrate, pH 7.2, Osmolarity: 255 mOsm). To characterize the ionic composition of currents, we performed ion substitution experiments or used specific blockers while recording with Tris-based internal saline. To block all $Na^+$ currents recordings were performed in NMDG-based extracellular saline solution (115 NMDG, 2 KCl, 5 HEPES, 10 glucose, 3 CaCl$_2$, 3 MgCl$_2$, pH 7.2, Osmolarity: 255 mOsm). Fast $Na^+$ currents were blocked by the addition of 1 μM or 30 μM TTX (Tocris Biosciences). Na$_v$1.6 channels were blocked by the addition of 5 μM MV1312 (4-Chloro-n-(3-[2-(4-methoxy-phenyl)-1 h-imidazol-4-YL]-phenyl)-benzamide, a gift from Dr. Mirko Rivara [Università di Parma, Parma, Italy] [Weuring et al, 2020]). Where lidocaine (1 μM; Sigma-Aldrich) was used to block $Na^+$ currents, recordings were preceded by a 10 s depolarizing step to 0 mV to allow lidocaine to bind the open Na$_v$ channels. To block $Ca^{2+}$ currents recordings were performed in the presence of 100 μM CdCl$_2$. To block the influx of $K^+$ ions recordings were performed in TEA-containing external saline (in mM: 65 NaCl$_2$, 4 KCl, 5 HEPES, 0.01 glycine, 10 glucose, 50 TEA-Cl, 3 CaCl$_2$, 3 MgCl$_2$, pH 7.2, Osmolarity: 250 mOsm).

## Behavior experiments

For behavior experiments, tadpole optic tectum was bulk electroporated with Na$_v$1.1 plus Na$_v$1.6 MOs or Control MO at stage 44–45 to perturb $Na^+$ channels when intrinsic excitability is highest during development (Pratt & Aizenman, 2007). Behavioral experiments were then carried out at stage 49 in the first 5 h of the light cycle when tadpoles are most active. For visual acuity and multisensory integration behavior experiments, individual tadpoles were placed in a 5-cm diameter dish filled with Steinberg's solution and illuminated by 4 IR lights placed uniformly to encircle the dish. Trials were recorded at 30 frames/s with a SCB-2001 HAD CCD camera (Samsung), with the presentation of stimuli on CRT monitor located beneath the dish controlled by custom MATLAB scripts (James et al, 2015; Truszkowski et al, 2017). Tracking was performed in real time using EthoVision XT (Noldus Information Technology), with tadpoles that did not move over a period of three consecutive minutes eliminated from the analysis.

## Visual acuity behavior

Individual tadpoles were exposed to a series of 32 block randomized stimulus presentations of counterphasing sine wave gratings of different spatial frequencies (3, 4.5, 9, and 18 cycles/cm) presented at 4 Hz for 4 s, with a 30-s inter-stimulus interval. Gratings were grey scale bars at 80% contrast, which have previously been shown to trigger robust escape responses (Schwartz et al, 2011; Truszkowski et al, 2017). For analysis, tadpole velocity was averaged over the 1 s pre-stimulus and during the stimulus, from which the absolute value of the change in velocity was calculated. The response of individual tadpoles to each spatial frequency was analyzed by calculating a Z-score, whereby the

mean response to each stimulus was compared with the mean response to no stimulus. Data shown represent individual trials from 27 Ctrl MO and 26 Na$_v$1.1 + Na$_v$1.6 MOs tadpoles from seven independent experiments. 3 Ctrl MO and 4 Na$_v$1.1 + Na$_v$1.6 MOs tadpoles were excluded from analysis because of inactivity. Data were compared using a two-way ANOVA with a Holm–Sidak correction for multiple comparisons.

### Multisensory integration behavior

Individual tadpoles were exposed to a series of 40 block randomized stimulus presentations of visual, mechanosensory or visual plus mechanosensory stimuli with inter-stimulus intervals of 0, 250, and 500 ms. Stimuli were presented for 2 s with an inter-stimulus interval of 30 s. Visual stimuli consisted of greyscale stripes of 25% contrast that alternated at 4 Hz, which has been shown to be sub-threshold. That is, the visual stimulus alone does not trigger a startle response in tadpoles (Truszkowski et al, 2017). Mechanosensory stimuli comprised low-volume clicks played through two speakers connected to the dish such that they vibrated the liquid of the dish, which has previously been shown to be subthreshold (James et al, 2015). For multisensory stimuli, visual stimuli were preceded by mechanosensory stimuli at 500, 250, and 0 ms interstimulus intervals (Fig 7D). To analyze the response of tadpoles to stimulus presentation, tadpole velocity was averaged over 1 s pre-stimulus and during the stimulus, from which the absolute value of the percent change in velocity was calculated. Values shown are normalized to the change in velocity to no stimulus, with all trials from each stimulus condition averaged. From these responses, we calculated the multisensory (MS) index ([multisensory – unisensory]/unisensory). Data shown were calculated with the visual stimulus used for the unisensory response; however, there was no difference if the mechanosensory stimulus was used. Data represent trials from 27 Ctrl MO and 25 Na$_v$1.1 + Na$_v$1.6 MOs tadpoles from seven independent experiments. 3 Ctrl MO and 5 Na$_v$1.1 + Na$_v$1.6 MO tadpoles were excluded from analysis because of inactivity. Data were compared using a one-way ANOVA with a Holm–Sidak correction for multiple comparisons.

### Schooling behavior

Schooling experiments were performed as recently described (Lopez et al, 2021), with experiments and analysis conducted using available code (https://github.com/khakhalin/Xenopus-Behavior). For each experiment, 30 Ctrl MO or Na$_v$1.1 + Na$_v$1.6 MOs tadpoles were transferred to a 17-cm diameter glass bowl on a LED tracing tablet (Picture/Perfect light pad), which in turn sat atop a dental vibrator (Jintai). Images of the bowl were captured using a GoPro Hero 7 (GoPro Inc.) every 5 min over the course of 1 h, with tadpoles dispersed by a vibration that was triggered 150 s before each image capture. For analysis, the position and heading of each tadpole was identified by acquiring the x–y coordinates of the head and gut of each tadpole using the multipoint tool in FIJI. A Delaunay triangulation was then used to calculate inter-tadpole distance and angles between neighboring tadpoles, which were then compared using a Kolmogorov–Smirnov test.

### Statistics and reproducibility

Statistics were performed in Prism 8 (GraphPad Software). Normally distributed data are presented as mean ± SD and analyzed using Welch's $t$ test or Welch's one-way ANOVA with a Dunnet T3 test for multiple comparisons. Nonparametric data is presented as median with IQR and analyzed with a Mann–Whitney $U$ test or a Kruskal–Wallis with Dunn's test for multiple comparisons. To measure the relationship between biophysical properties of stage 49 tectal neurons, a multivariate analysis was performed by calculating individual Pearson correlation coefficients between all biophysical properties measured with significance assessed by Bonferroni-corrected $P$-values. Sample sizes were based on power analyses and known biological variability from prior work (Pratt & Aizenman, 2007; Ciarleglio et al, 2015; James et al, 2015; Truszkowski et al, 2017). All experiments were performed with matched controls from the same clutch of embryos and from at least three separate breedings, with analysis blinded to the treatment group. Electrophysiology experiments were performed in a randomized manner, with analysis performed blinded to the identity of the treatment group. A small number of immobile animals were excluded from behavioral analyses as outlined in the relevant Materials and Methods sections. No outliers were excluded from the data analysis. It is not possible to distinguish sex at the developmental stages studied.

# Supplementary Information

# Acknowledgements

This work was supported by R01 EY027380 awarded to CD Aizenman from the National Eye Institute. We thank members of the CD Aizenman laboratory for their valuable intellectual input, and especially thank Kevin Keary for early contributions and Virgilio Lopez and Mimi Oupravanh for animal care. We are grateful to Dr. Mirko Rivara (Università di Parma, Parma, Italy) for the kind gift of the MV1312 that was used in this study.

### Author Contributions

AC Thompson: conceptualization, resources, data curation, formal analysis, validation, investigation, visualization, methodology, project administration, and writing—original draft, review, and editing.
CD Aizenman: conceptualization, data curation, formal analysis, supervision, funding acquisition, investigation, methodology, project administration, and writing—original draft, review, and editing.

### Conflict of Interest Statement

The authors declare that they have no conflict of interest.

**Life Science Alliance**

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
