## [Reviewer comments · Life Science Alliance]

Life Science Alliance

Characterization of Na⁺ currents regulating intrinsic excitability of optic tectal neurons

Adrian Thompson and Carlos Aizenman

DOI: <https://doi.org/10.26508/lsa.202302232>

Corresponding author(s): Carlos Aizenman, Brown University

Review Timeline:

Submission Date:	2023-06-22
Editorial Decision:	2023-08-14
Revision Received:	2023-10-11
Editorial Decision:	2023-10-13
Revision Received:	2023-10-20
Accepted:	2023-10-23

Transaction Report:

August 14, 2023

Re: Life Science Alliance manuscript #LSA-2023-02232

Dr. Carlos D Aizenman
Brown University
Neuroscience
Box G-LN
Brown University
Providence 02912

Dear Dr. Aizenman,

Thank you for submitting your manuscript entitled "Characterization of Na⁺ currents regulating intrinsic excitability of optic tectal neurons" to Life Science Alliance. The manuscript was assessed by expert reviewers, whose comments are appended to this letter. We invite you to submit a revised manuscript addressing the Reviewer comments.

Thank you for this interesting contribution to Life Science Alliance. We are looking forward to receiving your revised manuscript.

Sincerely,

B. MANUSCRIPT ORGANIZATION AND FORMATTING:

Reviewer #1 (Comments to the Authors (Required)):

This manuscript by Thompson and Aizenman examines the composition of channels responsible for Na currents in the neurons of the developing optic tectum in *Xenopus* tadpoles. By choosing a stage when intrinsic excitability is naturally evolving in response to the increase of afferent input, and when visual-stimulation induced plasticity of excitability can be easily activated, the authors present a methodical and systematic evaluation of the contributions of different Na currents to neuronal excitability. Not only do they identify novel TTX-insensitive channels associated with persistent and resurgent currents, but they show that Nav1.6 in particular plays a key role in activity-dependent regulation of intrinsic excitability. It is an impressive paper that takes advantage of a large number of powerful approaches, including novel pharmacology, knockdowns and behavioural assays. It is also extremely technically competently performed. Overall an impressive study.

My sole methodological concern in this paper is the validation of the MO constructs, which appear to be described as translation-blocking on the basis of their effects on Na currents, which is a bit circular logic given that that is what is being tested. However, because the current measurements are in fact a direct measure of channel expression and function, I believe the 'circularity' may be moot. If possible an experiment demonstrating specific knockdown of expression of a co-electroporated GFP-tagged NaV construct to reveal the specificity of the manipulation could strengthen the conclusions regarding specificity. However given the direct measurements of the currents by electrophysiological recordings, I consider this a suggestion and not a condition for acceptance.

Specific points:

1. Line 124 "between with"
2. Line 177 "where"
3. Line 256 "intrinsically excitability"
4. Line 297 This is one of the most important results in the paper, yet no traces are provided. Is there any possibility of showing example experimental traces of the lidocaine effect?
5. Line 323 "convey" "confer"
6. Line 366 remove "had"
7. Line 368 remove "has"
8. Line 378 - consider providing independent evidence for the specificity of the MO constructs.
9. Line 424,5,7 - It would be clearer if you put "MO" after each construct. Rather than saying "electroporated with Nav1.1 and Nav1.6MOs" say "electroporated with Nav1.1 MO and Nav1.6MO"
10. In the discussion it might be interesting for the authors to comment on the surprising speed with which new Na channels are translated, folded and inserted in the membrane. 4 hr of stimulation seems very fast.
11. One issue that is not mentioned, but which is likely highly relevant is the fact that in tectal neurons the axon initial segment emits from somewhere in the dendritic arbor. Thus the highest concentration of Nav1.6 channels is likely quite far away from the soma. Do the authors think this could impact their measurements or conclusions?
12. Line 626 - this is not a sentence.
13. Figure S1 - this table uses boldface to indicate significant Pearson correlations. However when such a large number of comparisons is being made, a Bonferroni correction for significance (not correlation) is in order.

Note on Reviewer #2: I largely concur with their conclusions and agree that additional discussion of the potential differences between maturational and activity-dependent channel expression would make a nice addition.

Reviewer #2 (Comments to the Authors (Required)):

In this study, Thompson and Aizenman have investigated the molecular mechanisms underlying the changes in intrinsic excitability in tectal neurons during development (stage42-49) and following sensory experience (induced by 4h of EVS). They show that changes in intrinsic excitability is strongly correlated with those in Nav1.1 and Nav1.6 expression, as well as in the amplitudes of all three types of Na⁺ current identified in these neurons (fast, persistent, and resurgent). Among these, Nav1.6

contributes to all types of Na⁺ current, whereas Nav1.1 is only important for the persistent and resurgent currents. They further show that pharmacologically blocking Nav1.6 attenuates basal excitability, and suppressing its expression abolishes EVS-induced increase in intrinsic excitability. Finally, they demonstrate that suppression of Nav expression during development has caused profound behavioral deficits, likely due to abnormal tectal circuit development.

Results from this study provide important insights on the type of ion channels that underlie the developmental and sensory regulation of intrinsic excitability in tectal neurons. The discovery that persistent and resurgent Na⁺ currents are insensitive to TTX is particularly interesting as these Na channels can be blocked by TTX in mammalian neurons. Perhaps most importantly, since the authors still observe profound behavioral deficits after Nav suppression when Na currents have already more or less gone back to normal, I think this can be a very useful model for neurodevelopmental disorders where behavioral phenotypes persist even though circuit dysfunction is no longer present. The results are clear and support the claims well. My only question is that, the correlation between intrinsic excitability and Na currents is convincing, but the Nav expression pattern does not look that congruent with the other two properties. Especially when excitability drops from stage 46 to 49, Nav expression levels either remain stable or increase. If intrinsic excitability is mediated by Na channels, then what could be causing this reduction in Na current amplitude and excitability? Along this line, 4 β expression has not changed during development but undergone substantial upregulation after EVS, so it seems to me that the mechanisms underlying developmental and sensory regulation of intrinsic excitability could be entirely different (given the role of 4 β in resurgent current). Could the authors address these in the Discussion?

Minor issues:

1. Figure S2 B, D, F; there are two y axes but only one type of data, how are the values normalized?
2. Non-sentence expressions at several places in the Methods section (Lines 701-702, 705-706, 708).
3. In the electrophysiology section under Methods, solution recipes should include the full name of chemicals used for others to replicate (e.g., ATP/GTP should be in their salt forms).

Response to reviewers

We thank the reviewers for their constructive comments that helped us to strengthen our analyses and more effectively communicate our findings and their significance. To address their comments, we have performed new statistical analyses, and revised the text and figures. We believe these changes have improved our manuscript. In this letter we address the specific points from the reviewers in detail and provide line numbers of important parts of the text that were changed.

Reviewer #1

This manuscript by Thompson and Aizenman examines the composition of channels responsible for Na currents in the neurons of the developing optic tectum in Xenopus tadpoles. By choosing a stage when intrinsic excitability is naturally evolving in response to the increase of afferent input, and when visual-stimulation induced plasticity of excitability can be easily activated, the authors present a methodical and systematic evaluation of the contributions of different Na currents to neuronal excitability. Not only do they identify novel TTX-insensitive channels associated with persistent and resurgent currents, but they show that NaV1.6 in particular plays a key role in activity-dependent regulation of intrinsic excitability. It is an impressive paper that takes advantage of a large number of powerful approaches, including novel pharmacology, knockdowns and behavioural assays. It is also extremely technically competently performed. Overall an impressive study.

My sole methodological concern in this paper is the validation of the MO constructs, which appear to be described as translation-blocking on the basis of their effects on Na currents, which is a bit circular logic given that that is what is being tested. However, because the current measurements are in fact a direct measure of channel expression and function, I believe the 'circularity' may be moot. If possible an experiment demonstrating specific knockdown of expression of a co-electroporated GFP-tagged NaV construct to reveal the specificity of the manipulation could strengthen the conclusions regarding specificity. However given the direct measurements of the currents by electrophysiological recordings, I consider this a suggestion and not a condition for acceptance.

As stated by the reviewer, MO constructs were validated based on their effects on Na⁺ currents and excitability given that these are a direct measure of Na⁺ channel expression and function, using the lissamine-tag that was conjugated to each MO to ensure that recordings were performed in tectal neurons that contained the MO constructs. Furthermore, the effect of MO constructs was observed following experience dependent suppression of excitatory synaptic inputs, a condition known to cause an increase in both intrinsic excitability and Na⁺ current amplitude, which are associated with increased Na⁺ channel density. While it would be of interest to determine the degree to which each MO construct reduces the expression of a co-electroporated GFP-tagged Na⁺ channel, we believe that the synthesis and characterization of a GFP-tagged channel for each of the Na⁺ channel subunits tested here are beyond the scope of our study.

Specific points:

Line 297 - This is one of the most important results in the paper, yet no traces are provided. Is there any possibility of showing example experimental traces of the lidocaine effect?

In Figure 4 we show the effect of blocking TTX-sensitive Na⁺ currents or all Na⁺ currents with NMDG-based external, which illustrated the surprising TTX-resistance of persistent and resurgent Na⁺ currents in tectal neurons. For brevity, we had chosen to omit lidocaine traces. However, we agree with the reviewer that the inhibitory effect of lidocaine on all Na⁺ currents is an important piece of evidence. We have therefore added Supplementary Figure 5 to illustrate the effect of lidocaine on fast, persistent and resurgent Na⁺ currents. The quantification of this effect remains located in Supplementary Table 2 [*See line 289 for in text reference to supplementary Fig 5 & 1327-1333 for the inserted figure legend*].

Line 378 - consider providing independent evidence for the specificity of the MO constructs.

We have addressed this point in our above response.

In the discussion it might be interesting for the authors to comment on the surprising speed with which new Na channels are translated, folded and inserted in the membrane. 4 hr of stimulation seems very fast.

Our results indicate that there is a correlation between increased Na⁺ channel expression and increases in Na⁺ current amplitude and excitability as a result of enhanced sensory experience, with Nav1.6 channels being particularly important for this process. While 4hrs of stimulation seems very fast, there is ample evidence from the literature to show the importance of protein synthesis for structural and functional changes in tectal neurons as they respond to as little as 30min of sensory experience. Nevertheless, this is an important point that warrants further discussion. As such, we have updated our discussion to include a more in-depth analysis of how Na⁺ channels may be regulated over 4 hours to elicit plasticity of excitability, and the possible contribution of newly expressed channels to this process [**See lines 642-686**].

One issue that is not mentioned, but which is likely highly relevant is the fact that in tectal neurons the axon initial segment emits from somewhere in the dendritic arbor. Thus the highest concentration of Nav1.6 channels is likely quite far away from the soma. Do the authors think this could impact their measurements or conclusions?

We restricted our measurements to peak current amplitude, and not activation and inactivation kinetics in order to eliminate possible space-clamp issues, consistent with previous studies (Aizenman et al., 2003; Pratt and Aizenman, 2007). Therefore, we believe that changes in Na⁺ currents that we measured in our experiments are due to changes in Na⁺ channel function, and not the result of a change in our ability to measure the maximal Na⁺ current amplitude. We have updated our results and discussion to more reflect this [**See lines 115-117 & 530-541**].

Figure S1 - this table uses boldface to indicate significant Pearson correlations. However when such a large number of comparisons is being made, a Bonferroni correction for significance (not correlation) is in order.

Our statistical analysis has been updated to include a Bonferroni correction for significance in our multivariate analysis of the biophysical properties of stage 49 tectal neurons. We have updated Supplementary Figure 1 and the associated figure legend to indicate that bolded values denote Pearson correlations that were significant following the Bonferroni correction for multiple comparisons. Because the stated correlation between the amplitude of the fast Na⁺ current and action potential threshold potential was no longer significant, we have removed this observation from the results section [**removed from line number 121**]. No other correlations discussed in our results were affected.

Note on Reviewer #2: I largely concur with their conclusions and agree that additional discussion of the potential differences between maturational and activity-dependent channel expression would make a nice addition.

We have updated and expanded our discussion to emphasize the importance of the differences observed between maturational and activity-dependent changes in Na⁺ channel expression and what functional implications there are for the regulation of Na⁺ currents and neuronal excitability in these two conditions [**See lines 593-641**].

Reviewer #2

In this study, Thompson and Aizenman have investigated the molecular mechanisms underlying the changes in intrinsic excitability in tectal neurons during development (stage42-49) and following sensory experience (induced by 4h of EVS). They show that changes in intrinsic excitability is strongly correlated with those in Nav1.1 and Nav1.6 expression, as well as in the amplitudes of all three types of Na⁺ current identified in these neurons (fast, persistent, and resurgent). Among these, Nav1.6 contributes to all types of Na⁺ current, whereas Nav1.1 is only important for the persistent and resurgent currents. They further show that pharmacologically blocking Nav1.6 attenuates basal excitability, and suppressing its expression abolishes EVS-induced increase in intrinsic excitability. Finally, they demonstrate that suppression of Nav expression during development has caused profound behavioral deficits, likely due to abnormal tectal circuit development.

Results from this study provide important insights on the type of ion channels that underlie the developmental and sensory regulation of intrinsic excitability in tectal neurons. The discovery that persistent and resurgent Na⁺ currents are insensitive to TTX is particularly interesting as these Na channels can be blocked by TTX in mammalian neurons. Perhaps most importantly, since the authors still observe profound behavioral deficits after Nav suppression when Na currents have already more or less gone back to normal, I think this can be a very useful model for neurodevelopmental disorders where behavioral phenotypes persist even though circuit dysfunction is no longer present. The results are clear and support the claims well.

My only question is that the correlation between intrinsic excitability and Na currents is convincing, but the Nav expression pattern does not look that congruent with the other two properties. Especially when excitability drops from stage 46 to 49, Nav expression levels either remain stable or increase.

We agree with the reviewer that the correlation between Na⁺ channel gene expression and intrinsic excitability/Na⁺ current amplitude is less striking as excitability and current amplitude decreases between developmental stages 46 and 49. While the reviewer points out that Na⁺ channel gene expression remains stable or increases, we believe that our data shows subtype specific differences in expression across development that are consistent with divergent functions for these Na⁺ channels in regulating homeostatic changes in excitability. Firstly, Na_v1.1 channel expression appears to increase across development from stage 42 to stage 49, suggesting that Na_v1.1 channel expression correlates with the maturation of tectal neurons, but plays a limited role in adapting Na⁺ currents as the excitability of tectal neurons changes. This would be consistent with both our observation that Na_v1.1 MO did not suppress the EVS-mediated increase in the fast Na⁺ current or excitability. Secondly, while Na_v1.6 channel expression was not significantly changed between stages 46 and 49, there was a trend downwards, as supported by Na_v1.6 channel expression no longer being significantly increased compared to stage 42. Furthermore, as evidenced by figure 1, while on average neuronal excitability is lower at stage 49 than stage 46, highly excitable cells can still be observed, which would fit with our observation that Na_v1.6 expression trends lower from stage 46 to stage 49. Given that the Na_v1.6 MO suppressed the EVS-mediated increase in the fast Na⁺ current or excitability, we believe that these data fit with our interpretation that changes in excitability of tectal neurons is largely mediated by changes in the expression of Na_v1.6 channels. We have updated our discussion to expand on these findings and possible interpretations of our data [*see line numbers 607-620*].

If intrinsic excitability is mediated by Na channels, then what could be causing this reduction in Na current amplitude and excitability? Along this line, 4β expression has not changed during development but undergone substantial upregulation after EVS, so it seems to me that the mechanisms underlying developmental and sensory regulation of intrinsic excitability could be entirely different (given the role of 4β in resurgent current). Could the authors address these in the Discussion?

We have updated our discussion to include a discussion of potential mechanisms that may mediate the homeostatic decrease in excitability in tectal neurons, including mechanisms that would act to remove Na⁺ channels from the membrane as well as mechanisms such as translational control that would control Na⁺ channel expression [*see lines 687-703*]. We have also added discussion on the possible roles for the accessory protein Na_v4β in the regulation of Na⁺ channel function in mature neurons versus developing tectal neurons [*see lines 621-630*]. These updates are included in an expanded discussion relating to the importance of the differences observed between maturational and activity-dependent changes in Na⁺ channel expression and what functional implications there are for the regulation of Na⁺ currents and neuronal excitability in these two conditions [*See lines 593-641*].

Minor issues:

Figure S2 B, D, F; there are two y axes but only one type of data, how are the values normalized?

We have updated the figure legend for Supplementary Figure S2 to clarify that total resurgent Na⁺ current was calculated by measuring the charge (area under the curve) over the repolarizing resurgent step, and that this is plotted on the right y-axis whereas individual currents are plotted on the left y-axis [*See lines 1273-1292*].

In the electrophysiology section under Methods, solution recipes should include the full name of chemicals used for others to replicate (e.g., ATP/GTP should be in their salt forms).

We thank the reviewers for pointing out this omission. Our methods have been updated to include the full name of ATP and GTP salts *[see line number 810]*.

October 13, 2023

RE: Life Science Alliance Manuscript #LSA-2023-02232R

Dr. Carlos D Aizenman
Brown University
Neuroscience
Box G-LN
Brown University
Providence 02912

Dear Dr. Aizenman,

Thank you for submitting your revised manuscript entitled "Characterization of Na⁺ currents regulating intrinsic excitability of optic tectal neurons". We would be happy to publish your paper in Life Science Alliance pending final revisions necessary to meet our formatting guidelines.

- please add ORCID ID for the corresponding author--you should have received instructions on how to do so
- please note that titles in the system and on the manuscript file must match
- abstract should be no more than 175 words
- please separate the Figure Legends and Supplemental Figure Legends into separate sections
- we encourage you to revise the figure legend for Figure 6 such that the figure panels are introduced in an alphabetical order
- please add callouts for Figures S3A-C; S4A-D; 7I; S5A-B and table S1 to your main manuscript text

A. FINAL FILES:

B. MANUSCRIPT ORGANIZATION AND FORMATTING:

Sincerely,

October 23, 2023

RE: Life Science Alliance Manuscript #LSA-2023-02232RR

Dr. Carlos D Aizenman
Brown University
Neuroscience
Box G-LN
Brown University
Providence 02912

Dear Dr. Aizenman,

Thank you for submitting your Research Article entitled "Characterization of Na⁺ currents regulating intrinsic excitability of optic tectal neurons". It is a pleasure to let you know that your manuscript is now accepted for publication in Life Science Alliance. Congratulations on this interesting work.

DISTRIBUTION OF MATERIALS:

Again, congratulations on a very nice paper. I hope you found the review process to be constructive and are pleased with how the manuscript was handled editorially. We look forward to future exciting submissions from your lab.

Sincerely,
